# When to Use Which? An Investigation of Search Methods on Expensive Black-box Optimisation Problems

## Abstract

Many real-world optimisation problems are black-box in the sense that the structure of their objective function is not accessible or exploitable. Some of such Black-Box Optimisation (BBO) problems are also expensive, thanks to the use of simulations, experiments or costly computations to evaluate a solution (i.e., calculate its objective function value). Despite the prevalence of expensive BBO, different practical scenarios may require different computational resources and search budgets. In some scenarios, evaluating a solution may take hours or days (e.g., in drug design), allowing generating only a few hundred solutions at most, while in some other scenarios, the budget is more generous in which evaluating a solution takes a couple of minutes (e.g., in software configuration tuning), hence allowing a few thousand solutions to be generated. Consequently, a relevant question is that among various popular search methods for BBO (e.g., Bayesian optimisation and evolutionary algorithms), which one is the first choice for practitioners to use under different levels of tightness of their budget, and also what if some domain knowledge of the problem (e.g., ruggedness level of the search space) is available.

In this paper, we aim to answer these questions. Through an extensive experimental study on a suite of test functions with various features, we observe that some methods which were believed unsuitable for expensive BBO are actually competitive under certain circumstances; for example, Nelder Mead on small-size problems with simple landscapes under fairly tight budgets (e.g., 200–800 evaluations) and CMA-ES on medium-sized problems under fairly generous budgets (e.g., $\geq 800$). On the other hand, Bayesian optimisation methods perform consistently well under very tight budgets (e.g., $\leq 200$) regardless of problem attributes and characteristics.

## 1 Introduction

The Black-Box Optimisation (BBO) problem refers to a class of problems where the structure of the objective function and/or the constraints defining the feasible space is unknown, inaccessible, or unexploitable. In many practical cases, the evaluation process of BBO (i.e., calculating the objective function of a given solution) is expensive or resource-intensive due to the use of simulations, experiments or costly computations, allowing a search algorithm only to generate/draw a limited number of solutions/samples (Jones et al., 1998; Shahriari et al., 2015; Garnett, 2023).

Despite falling into the same category of expensive BBO, different optimisation scenarios may require different amounts of time. There exist some practical scenarios having very tight budget constraints, taking hours (e.g., engine design (Ahrari et al., 2021)) or even days (e.g., molecular design (Li et al., 2024)) to evaluate a solution, hence only a few hundred solutions can be generated at most. On the other hand, there may be many scenarios with relatively more generous budgets, in which the evaluation process for a solution takes one or several minutes (e.g., software configuration tuning (Chen & Li, 2021)), thereby allowing a few thousand solutions to be generated.

It is known that Bayesian Optimisation (BO) (Kushner, 1964) is often regarded as the first choice to tackle expensive BBO problems. BO has been used in various application scenarios (Wang & Dowling, 2022; Binois & Wycoff, 2022; Huang et al., 2024; Wang et al., 2016), but one may want to know what its exact comfort zone is; for example, is it still the best choice when a relatively

generous budget is accessible (e.g., around 1,000 evaluations)? Some advanced variants of BO (e.g., TuRBO (Eriksson et al., 2019)) may work better under a larger number of evaluations (Ament et al., 2023). But, how do they perform compared to some evolutionary search methods such as Covariance Matrix Adaptation Evolution Strategy (CMA-ES) (Hansen & Ostermeier, 2001), which has recently been found to be effective in dealing with expensive BBO problems (Hutter et al., 2013; Andersson et al., 2015; Ozaki et al., 2022). In addition, in contrast to CMA-ES that considers an explicit probability model, what about other evolutionary computation (EC) techniques based on an implicit probability model like Genetic Algorithm (GA) (Holland, 1992) and Differential Evolution (DE) (Storn & Price, 1997), particularly when the computational resource is more relaxed (e.g., a couple of thousands of evaluations)? Besides, there also exist some other classical optimisation techniques which have been frequently used on expensive BBO problems such as Nelder Mead (NM) (Nelder & Mead, 1965). Are they competitive with BO and EC-based optimisers? If so, under what circumstances?

In addition, despite the fact that in BBO the objective function is not accessible, one may have some knowledge of the problem in practical optimisation scenarios. For example, in software configuration tuning, the fitness landscape is usually very rugged and the problem has a lot of local optima (due to the discretisation of decision variables) (Chen & Li, 2021). As such, a useful question is that when some domain knowledge of an expensive BBO problem (e.g., separability, multimodality) is available, what kind of search method is recommended to use.

Given the above, in this paper we aim to answer two questions: 1) under different levels of tightness of search budget in expensive BBO, what is the first optimisation algorithm to try in general? And 2) when some domain knowledge of the problem is available, what kind of algorithm is recommended to use then? We hope that answering these two questions would provide some guidance for practitioners to select suitable optimisation algorithms when encountering an expensive BBO problem in their field.

It is worth noting that empirical comparisons of search algorithms on BBO are not uncommon in the literature. A variety of studies have been conducted, each with a specific focus. These include: 1) investigation into the effect of algorithm configurations within particular classes of methods (Wang et al., 2019; Pošík & Klemš, 2012; Qin & Li, 2013; Varelas et al., 2018; Takenaga et al., 2023), such as the influence of acquisition functions in BO (Rehbach et al., 2020; Leite Richardson et al., 2024), or population size in evolutionary algorithms (Roeva et al., 2015; Piotrowski, 2017); 2) performance comparisons of search algorithms under specific optimisation scenarios (Vesterstrom & Thomsen, 2004; Panduro et al., 2009; Hansen et al., 2010; Pošík et al., 2012; Pošík & Kubalík, 2012; Lilla et al., 2013; Lim & Haron, 2013; Deb et al., 2014; Turner et al., 2021; Alibrahim & Ludwig, 2021; Ozaki et al., 2022; Raponi et al., 2023), such as in low-dimensional (Stripinis et al., 2025) and high-dimensional search spaces (Varelas et al., 2020; Santoni et al., 2024); 3) algorithm comparisons aimed at supporting automated algorithm selection (Muñoz et al., 2015; Kerschke et al., 2019; Yuen et al., 2019; Kerschke et al., 2019; Kerschke & Trautmann, 2019; Meunier et al., 2022). Among these, some studies do consider different budgets (Hutter et al., 2013; Loshchilov & Hutter, 2016; Raponi et al., 2023; Stripinis et al., 2025; Meunier et al., 2022). However, to the best of our knowledge, no existing work systematically compares algorithm performance across varying budgets, offers detailed recommendations to different budget levels, or accounts for the presence or absence of domain knowledge in the optimisation process.

## 2 PRELIMINARIES

It is not uncommon for derivative information on real-world problems to be either inaccessible, unreliable, or impractical (Golovin et al., 2017; Alarie et al., 2021; Meunier et al., 2022). Black-Box Optimisation (BBO), without using derivative information, aims at finding high-quality solutions for a given optimisation problem. Without loss of generality, let us consider a minimisation problem:

$$\min_{x \in \mathcal{X}} f(x) \tag{1}$$

where $x \in \mathcal{X}$ denotes variables in decision space, $\mathcal{X} \subseteq \mathbb{R}^d$ denotes a compact set, and $f$ is a black-box function.

To deal with BBO problems, a variety of methods have been developed, such as evolutionary algorithms, local search and Bayesian Optimisation (BO). Evolutionary algorithms are a large class of

nature-inspired global search methods, including GA (Holland, 1992), DE (Storn & Price, 1997) and CMA-ES (Hansen & Ostermeier, 2001). They have been demonstrated to be effective on many hard BBO problems like those having multiple local optima and rugged, deceptive landscapes (Muñoz et al., 2015; Marín, 2012). There also exist several local search algorithms, such as Nelder Mead (NM) (Nelder & Mead, 1965) which has been used for decades to solve numerous real-world problems due to its effective performance (Ozaki et al., 2017). Evolutionary and local search methods typically require large computational resources. When it comes to solving an expensive BBO problem, BO (Kushner, 1964) is usually believed to be among the first choices (Wang et al., 2018; Nayebi et al., 2019; Wang et al., 2023; Raponi et al., 2023; Jiang & Li, 2025a). It has been widely used in tackling expensive BBO problems in various domains such as automated machine learning (Galuzzi et al., 2020; Turner et al., 2021), chemical product design (Wang & Dowling, 2022), and robotics (Lizotte et al., 2007).

However, in some practical scenarios, the computational resources for expensive BBO problems may not be strictly tight; for example, it only takes around one minute to evaluate a solution in many configurable software systems (Chen & Li, 2021) or there are some high-performance facilities, enabling parallel computing (Garland et al., 2008). This allows a relatively large number of solutions to be considered during the search process (e.g., a few thousand) (Nishihara & Nakata, 2024; Nabae & Fukagata, 2021). Consequently, one may ask under such a more generous search budget, is BO still competitive compared with evolutionary and local search methods.

On the other side, in the literature there have been empirical, yet seemingly inconsistent, observations reported regarding the performance of search methods for expensive BBO problems. For example, Loshchilov & Hutter (2016); Santoni et al. (2024); Raponi et al. (2023) show that BO outperforms CMA-ES when the evaluation budget is limited, whereas Ozaki et al. (2022) claim that CMA-ES and NM outperform BO, especially in scenarios involving parallel evaluations. In Wang et al. (2019), their results show that BO can capture a high-quality solution with a smaller budget and GA can determine global best solutions with a larger budget, while Alibrahim & Ludwig (2021) show that the GA performs better than BO for hyper-parameter optimisation. Moreover, it has been found in some studies that evolutionary and local search methods are also suited to dealing with expensive BBO problems, e.g., GAs (Yuen et al., 2019) and NM (Takenaga et al., 2023).

These mixed findings indicate that different algorithms excel under varying conditions, but the circumstances under which one algorithm surpasses another remain unclear. Therefore, one may be curious about if evolutionary or local search algorithms can compete with Bayesian optimisation algorithms on some expensive BBO problems; if so, under what circumstances? And more practically, under different tightness levels of computational resources, what is the first optimisation algorithm for a practitioner to try in general?

To answer these questions, we consider six representative algorithms from the three classes of optimisation methods: BO, evolutionary and local search. They are vanilla BO (Kushner, 1964), TuRBO (Eriksson et al., 2019), GA (Holland, 1992), DE (Storn & Price, 1997), CMA-ES (Hansen & Ostermeier, 2001), and NM (Nelder & Mead, 1965). We consider vanilla BO as it is the most canonical BO algorithm, widely used in various expensive problems (Garnett, 2023; Wang et al., 2023). We also include TuRBO, a popular BO variant (Cowen-Rivers et al., 2022; Ament et al., 2023), particularly suitable for high-dimensional problems (Santoni et al., 2024; Xu et al., 2025). We consider three classical evolutionary algorithms, i.e., GA, DE, and CMA-ES, and one classical local search heuristic, i.e., NM. These algorithms have also been found to perform well in expensive BBO scenarios (Alibrahim & Ludwig, 2021; Ozaki et al., 2022; Takenaga et al., 2023). In addition, we also include Random Search (RS) (Karnopp, 1963) as a baseline. Detailed descriptions of these algorithms can be found in Appendix A.1.

It is worth noting that in this work we only consider classical methods representing different classes of optimisation techniques, which practitioners typically consider to employ when facing an expensive BBO problem in their applied fields. Hence, we do not include complicated, composite algorithms developed recently (Kumar et al., 2017; Jiang & Li, 2025b), such as those combining classical algorithms with surrogates (Mallipeddi & Lee, 2015; Bajer et al., 2019) or integrating multiple classical algorithms (Rapin & Teytaud, 2018). We also exclude algorithms designed to address specific issues in optimisation (Papenmeier et al., 2025; Xu et al., 2025), such as BBO problems with mixed search spaces (Daulton et al., 2022; Papenmeier et al., 2023) or cost-aware optimisation (Foumani et al., 2023; Xie et al., 2024).

## 3 Experimental Design

### 3.1 Test Problems

We consider the problem suite BBOB (Hansen et al., 2009) to test the algorithms. BBOB is a set of well-established functions with rich features, such as separability/non-separability, uni-modality/multi-modality, weak/moderate/high conditioning, and weak/adequate global structure. Compared to other benchmark functions, they have been found to well represent the variety of real-world scenarios (Long et al., 2022; Brockhoff et al., 2022; Santoni et al., 2023; 2024; Liang et al., 2024).

### 3.2 General Experimental Settings

Since we would like to see the performance of the algorithms under different budgets, we consider the entire search process up to 10,000 evaluations. In particular, we mainly discuss four tightness levels of search budgets: very tight (200 evaluations and below), fairly tight (200–800), fairly generous (800-2,000), and very generous (above 2,000). Note that for vanilla BO, the $O(n^3)$ time complexity of Gaussian processes, where $n$ is the number of samples, makes computations increasingly time-consuming (Williams & Rasmussen, 2006). For instance, one run of the vanilla BO on a 10-dimensional problem under a budget of 1,000 evaluations takes approximately four hours. Given that each instance requires 30 independent runs and the computational complexity increases rapidly, we cap the budget of vanilla BO at 1,000 evaluations to maintain computational efficiency. When the search budget exceeds 1,000, the best solutions identified by vanilla BO under 1,000 evaluations are compared with those obtained by the other methods under 1,000–10,000 evaluations. In addition, CMA-ES and NM may terminate before reaching 10,000 evaluations; for such a case, we restart the algorithm, following the common practice (Loshchilov, 2013; Hansen, 2009). For vanilla BO and TuRBO, we set the number of initial solutions to 2D for training an initial Gaussian process model according to (De Ath et al., 2021); Note that there is no observed drawback to starting with a small number of initial solutions (Forrester et al., 2008). All the algorithms use their commonly used/recommended settings; details are provided in Appendix A.1 due to space constraints.

In the experiments, we conduct 30 independent runs of each algorithm for the same instance of all the 24 BBOB functions, with the bounds normalised to the range [0,1]. We use the Wilcoxon rank-sum test (Wilcoxon, 1992) with Holm-Bonferroni correction (Holm, 1979) to determine if two methods are statistically different. The code and data are available at an anonymous repository[1].

## 4 Experimental Results

In this section, we first show general comparative results of the considered algorithms on all BBOB functions with a moderate number of dimensions (10) under varying budgets in Section 4.1. We then investigate the effect of different problem dimensionalities on the algorithms in Section 4.2. Afterwards, in Section 4.3, we show how the algorithms perform when some domain knowledge of optimisation problems is accessible, where the 24 functions are categorised into five groups according to their characteristics, following the classification by (Hansen et al., 2009). Finally, we summarise the results observed in Section 4.5.

### 4.1 Performance Analysis of the Algorithms under Varying Evaluation Budgets at Moderate Dimensionality

Table 1 shows the number of functions where each algorithm is statistically the best out of the 24 BBOB functions under the four budget levels (i.e., with 200, 800, 2,000 and 8,000 evaluations). Detailed results are provided in Tables 3–6 in Appendix B.1. As can be seen from the table, TuRBO and vanilla BO have a clear advantage under the very tight budget (200 evaluations). CMA-ES is in general the best for the other three situations, though the two BO algorithms are competitive on some problems when the budget is fairly tight (800 evaluations) and GA is competitive on a small fraction of problems when the budget is very generous (8,000 evaluations).

---

[1]https://anonymous.4open.science/r/W2W-E811

Table 1: The number of functions where each of the seven algorithms, Random Search (RS), Nelder Mead (NM), Genetic Algorithm (GA), Differential Evolution (DE), CMA-ES, vanilla BO and TuRBO, performs statistically the best out of the 24 BBOB functions over 30 independent runs, under the four budget tightness levels.

| Search budget | RS | NM | GA | DE | CMA-ES | Vanilla BO | TuRBO |
|---|---|---|---|---|---|---|---|
| 200 evaluations | 0 | 1 | 0 | 0 | 1 | 10 | 14 |
| 800 evaluations | 0 | 3 | 1 | 0 | 18 | 7 | 5 |
| 2,000 evaluations | 0 | 3 | 3 | 1 | 20 | 3 | 1 |
| 8,000 evaluations | 0 | 3 | 5 | 2 | 18 | 1 | 1 |

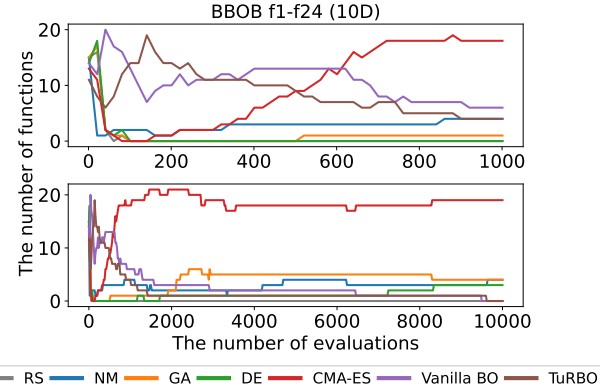

Figure 1: Trajectories of the number of functions where each algorithm is statistically the best out of the 24 BBOB functions over 30 independent runs, along the varying budget of up to 1,000 evaluations (top panel) and 10,000 evaluations (bottom panel).

To understand the performance of the comparative algorithms during the entire search process (i.e., under constantly varying search budget), Figure 1 shows the trajectories of the number of functions where each algorithm is statistically the best out of the 24 BBOB functions over 30 independent runs, along the varying budget of up to 1,000 evaluations (top panel) and 10,000 evaluations (bottom panel). Comparing vanilla BO and TuRBO, it is clear that when under around 100 evaluations, vanilla BO is better, while it is overtaken by TuRBO when a slightly relaxed budget is available (i.e., 100–200 evaluations). When the search budget reaches around 600 evaluations, CMA-ES catches up and outperforms the two BO algorithms, and after that, CMA-ES shows a clear advantage over the other algorithms until the maximum budget (10,000 evaluations) is reached. Considering the remaining algorithms, GA takes the second place when a more generous budget is available (above 2,000 evaluations). NM mostly ranks third along the entire search process, i.e., after vanilla BO and TuRBO when the budget is tight and after CMA-ES and GA when the budget is generous. In contrast, random search unsurprisingly performs worst, with its trajectory of zero throughout the whole process.

To help further understand the search behaviour of these algorithms, we consider their convergence trajectories on a representative test function, BBOB f9, shown in Figure 2. Convergence trajectories on the other BBOB functions can be found in Appendix B.2. The convergence trend of each algorithm on the function is generally consistent with the overall performance illustrated in Figure 1. As seen, vanilla BO performs best with very tight budgets (e.g., $\leq 100$ evaluations) but is then overtaken by TuRBO. When the budget reaches around 500 evaluations, CMA-ES surpasses both BO algorithms and maintains the first place until it reaches 10,000 evaluations, despite having a considerably higher standard deviation compared with the other algorithms.

## 4.2 EFFECT OF PROBLEM DIMENSIONALITY

In the last section, we have seen the comparative results on the BBOB functions with a moderate number of dimensions (i.e., 10D). Here, we want to see whether and how the problem dimensionality

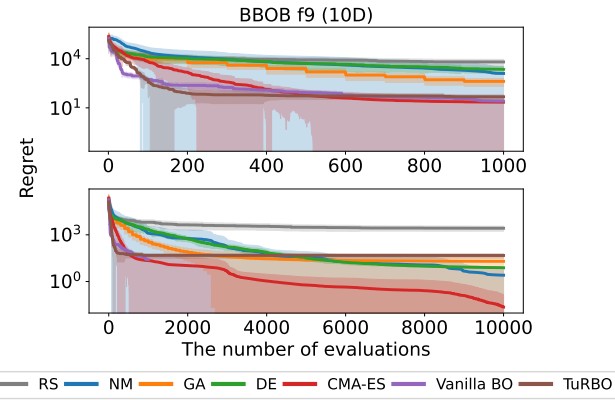

Figure 2: Convergence trajectories of the seven algorithms throughout the search process on a representative test function, BBOB f9. Here, each coloured line represents the mean regret (the difference between the true optimum and the best function value) obtained over 30 independent runs.

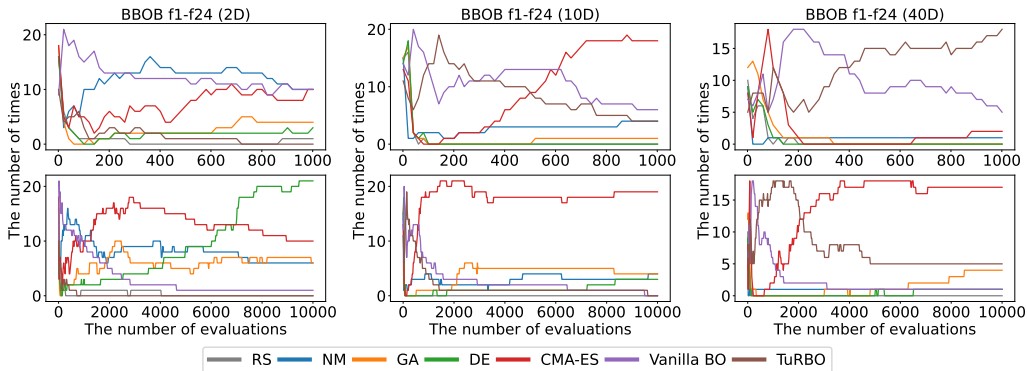

Figure 3: Trajectories of the number of functions each algorithm is statistically the best out of the 24 BBOB functions along the varying budget of up to 1,000 evaluations (top panel) and 10,000 evaluations (bottom panel), under 2, 10 and 40 problem dimensions.

affects the algorithm performance. Specifically, we consider two rather edge cases, where the problem dimensionality is 2D and 40D, respectively.

Figure 3 shows the trajectories of the number of functions on which each of the seven algorithms is statistically the best along the varying budget of up to 1,000 evaluations (top panel) and 10,000 evaluations (bottom panel), under 2, 10 and 40 dimensions. Firstly, considering the low-dimensional problems (i.e., 2D), interestingly, the picture (left of Figure 3) is pretty different from what we have seen on the 10D problems (centre of Figure 3). Apart from vanilla BO which still performs best under a very tight budget (200 evaluations), the algorithm NM is the best one under a fairly tight budget (200–1,000). One possible reason is that the simplex (a triangle in 2D here) used in NM can adapt its size and shape more effectively to fit the landscape of the objective function in low dimensions (Lagarias et al., 1998; Conn et al., 2009). When a more generous budget is available, CMA-ES performs best (1,000–7,500 evaluations), whereas DE becomes the best performer from around 7,500 evaluations. This is interesting, since DE constantly performs very poorly on the 10D problems (centre of Figure 3). This is consistent with the observations in the literature (Cai et al., 2019; Deng et al., 2021; Liu et al., 2023) that when applying classical DE to high-dimensional problems, the performance tends to worsen. In addition, it is unsurprising to see that TuRBO is among the very worst algorithms since it is designed for high-dimensional problems, hence not a good competitor under low-dimensional search spaces.

Consider the high-dimensional problems (40D) on the right-hand side of Figure 3. TuRBO shows a clear advantage between 400 and 2,000 evaluations. CMA-ES again is the best performer from

2,000 evaluations. Interestingly, vanilla BO, which was believed not suitable for high-dimensional search space (Frazier, 2018; Song et al., 2022; Chen et al., 2024), is very competitive under 1,000 evaluations, which is echoed by the results in Hvarfner et al. (2024); Papenmeier et al. (2025); Xu et al. (2025). And it even performs best between around 100 and 400 evaluations. Additionally, compared with BO algorithms, the other optimisers do not perform well on high-dimensional problems when the budget is tight.

### 4.3 What if Some Problem Characteristics Are Accessible

In dealing with a practical black-box optimisation problem, sometimes, some domain knowledge (e.g., landscape smoothness and multi-modality) may be known. For example, in software configuration tuning, it is known that the fitness landscape is usually very rugged and the problem has a lot of local optima (Chen & Li, 2021). In this section, we investigate the performance of the algorithms under different problem characteristics. Following the suggestion in Hansen et al. (2009), we categorise the 24 BBOB functions into 5 groups based on their characteristics. They are

- Functions with separate variables (f1-f5);

- Functions with low or moderate conditioning (mild slope) and few or no local optima (f6-f9);

- Functions with high conditioning (steep slope) and few or no local optima (f10-f14);

- Multi-modal functions with adequate global structure (f15-f19);

- Multi-modal functions with weak global structure (f20-f24).

Figure 4 shows the trajectories of the number of functions where each of the seven algorithms is statistically the best on problem categories with different characteristics. For functions with separate variables (f1–f5, Figure 4(a)), vanilla BO performs best with limited evaluation budgets across all the dimensionalities. TuRBO excels when the budget is between 400 and 2,000 evaluations, but only in the high-dimensional case. CMA-ES consistently performs best under a moderate search budget, regardless of problem dimensionality. It is worth mentioning that GA can effectively handle such variable-separate functions and works well across the three problem dimensionalities when the budget is sufficient.

For functions with low or moderate conditioning and few or no local optima (f6–f9, Figure 4(b)), NM performs best on the low-dimensional cases when the search budget is between 200 and 800 evaluations. CMA-ES shows overwhelmingly better performance when the budget exceeds 2,000, making it the best choice for all 9 functions with low and medium dimensionalities. Interestingly, vanilla BO outperforms TuRBO on the high-dimensional cases when the budget is between 200 and 400 evaluations. This suggests that vanilla BO remains effective on high-dimensional problems with simple landscapes.

Similar observations have been obtained on the functions with high conditioning and few or no local optima (f10–f14, Figure 4(c)). As can be seen in the figure, NM works well on the low-dimensional cases between around 150 and 2,000 evaluations, and CMA-ES shows a clear advantage when the budget exceeds 2,000. A slight difference from the last group (f6–f9) is on the 2D case, where DE becomes the best performer when sufficient computational resources are available.

For multi-modal functions with adequate global structure (f15–f19, Figure 4(d)), vanilla BO shows very good performance across all three dimensionalities with tight evaluation budgets. This again demonstrates that vanilla BO is competitive on high-dimensional problems, which is echoed by Papenmeier et al. (2025). Interestingly, CMA-ES's performance is not as good as on functions f1–f14 under low dimensionality. This may be due to the algorithm's focus on local search (Loshchilov, 2013), which was confirmed in Omidvar & Li (2010).

For the multi-modal functions with weak global structure (f20–f24, Figure 4(e)), CMA-ES is also not competitive on the 2D case. Vanilla BO does not perform very well on the medium-/high-dimensional cases, compared to its results on f1–f19. In contrast, TuRBO demonstrates excellent performance on the medium-/high-dimensional cases.

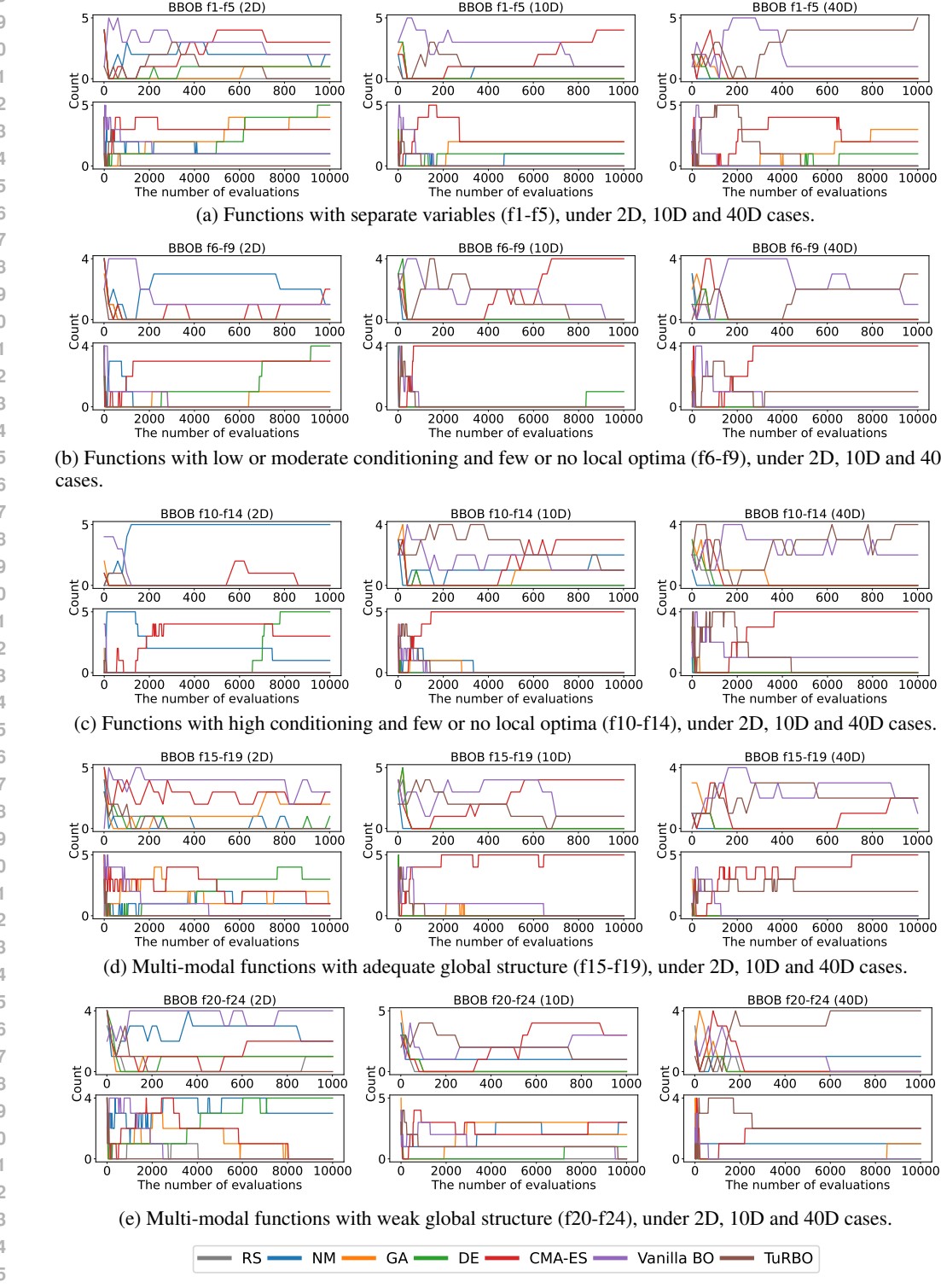

(a) Functions with separate variables (f1-f5), under 2D, 10D and 40D cases.

(b) Functions with low or moderate conditioning and few or no local optima (f6-f9), under 2D, 10D and 40D cases.

(c) Functions with high conditioning and few or no local optima (f10-f14), under 2D, 10D and 40D cases.

(d) Multi-modal functions with adequate global structure (f15-f19), under 2D, 10D and 40D cases.

(e) Multi-modal functions with weak global structure (f20-f24), under 2D, 10D and 40D cases.

Figure 4: Trajectories of the number of functions each algorithm is statistically the best on each of the five problem groups along the varying budget of up to 1,000 evaluations (top panel) and 10,000 evaluations (bottom panel) for different problem dimensionalities.

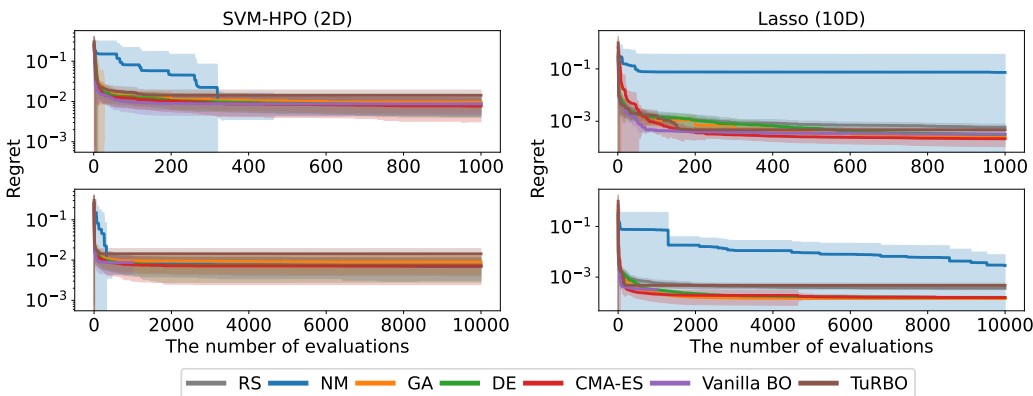

Figure 5: Convergence trajectories of the seven algorithms throughout the search process on the SVM-HPO problem (2D) (Eggensperger et al., 2021) and Lasso problem (10D) (Šehić et al., 2022). Here, each coloured line represents the mean regret (the difference between the true optimum and the best function value) obtained over 30 independent runs.

### 4.4 ON REAL-WORLD PROBLEMS

Although the BBOB function benchmark is widely regarded as representatives of the typical difficulties that arise in real-world applications (Pál et al., 2012), it is useful to validate whether the performance trends observed on them also hold on practical optimisation scenarios. Here, we consider two real-world continuous black-box problems in hyperparameter optimisation, the SVM-HPO problem from HPOBench (Eggensperger et al., 2021) and the Lasso problem from LassoBench (Šehić et al., 2022). The former one has two variables, aiming to minimise the validation loss, defined as (1 - accuracy). The latter has 10 variables and is to minimise the mean squared error (MSE).

Figure 5 shows convergence trajectories of the seven algorithms throughout the search process on the two real-world problems. As can be seen from the figure, the performance of the optimisation methods on these real-world problems is in general consistent with their performance on the BBOB benchmark. When the budget is very tight ($\leq 200$), vanilla BO (purple line) performs best. As the budget increases, CMA-ES (red line) catches up, overtakes vanilla BO, and remain the top-performing method thereafter. It is worth noting that, interestingly, on the 10D Lasso problem, GA becomes slightly better than CMA-ES once the budget exceeds 2,000. A possible explanation is that, in the Lasso task, only 3 of the 10 variables substantially interact and contribute to the objective value, making the problem largely variable-separable (Šehić et al., 2022).

### 4.5 SUMMARY

Table 2: The algorithm(s) to try when facing an expensive BBO problem, under the different budget levels and problem dimensionalities. When more than one algorithm is recommended, they are listed in order of priority from left to right.

| Budget (evaluations) | Low-dimensional | Medium-dimensional | High-dimensional |
|---|---|---|---|
| Very tight (e.g., $\leq$200) | Vanilla BO | Vanilla BO/TuRBO | Vanilla BO |
| Fairly tight (e.g., 200–800) | NM/vanilla BO | Vanilla BO/TuRBO/CMA-ES* | TuRBO/vanilla BO |
| Fairly generous (e.g., 800–2,000) | CMA-ES | CMA-ES | TuRBO |
| Very generous (e.g., $\geq$2,000) | CMA-ES/DE** | CMA-ES | CMA-ES |

\* CMA-ES becomes the best when the budget exceeds 600.
\*\* DE becomes the best when the budget exceeds 8,000.

Table 2 sums up the results observed. As can be seen from the table, when the search budget is very tight (only $\leq$200 evaluations allowed), vanilla BO is always the first choice. When a fairly tight budget is available (200–800), vanilla BO can be a good choice, but in low-dimensional cases (e.g., 2D), the local search algorithm NM may be better, and in high-dimensional cases (e.g., 40D), TuRBO

can be the first one to try. When there is a fairly generous budget (800–2,000), CMA-ES is the one recommended in low and medium-dimensional cases, while TuRBO is the one in high-dimensional cases. Under the situation that the budget is very generous ($\geq$2,000), CMA-ES is always the one to trust, except in the low-dimensional case where DE is recommended when more budget is available (from around 8,000 evaluations).

Under the circumstances that some problem characteristics are accessible, the recommendation of the algorithms can be further summarised to the following.

- Random search is never recommended to use under any circumstances.

- NM is a good choice on low-dimensional problems with simple fitness landscapes, e.g., uni-modal structure.

- GA can be used on problems with separate variables, regardless of the dimensionalities, when a sufficient budget is available (e.g., 6,000 evaluations or more).

- DE is the one to recommend on low-dimensional problems, regardless of the problem characteristics, under a sufficient budget (e.g., 7,000 evaluations or more).

- CMA-ES is always the best provided that the budget is not too tight, but its advantage becomes less apparent when the problem's fitness landscape is complex (e.g., multi-modal and/or with weak global structure).

- Vanilla BO is the best choice under a tight budget, particularly for problems with simple fitness landscape (e.g., low or moderate conditioning or adequate global structure), even in a high-dimensional space (e.g., 40D).

- TuRBO can be a better choice than vanilla BO on complex problems (e.g., with weak global structure), when a medium number of variables are involved (e.g., $\geq$10).

## 5 CONCLUSION

In this work, we have conducted an extensive experimental study to compare the performance of several popular optimisation methods (including BO, CMA-ES, and other evolutionary and local search algorithms) for expensive BBO problems, under different levels of tightness of search budget. We make some recommendations for the use of those algorithms under different conditions of budget, problem dimensionality, and problem characteristics; the last one is particularly for the case that some domain knowledge is accessible. Note that in this study we mainly consider "classical" BBO algorithms, serving the purpose that people who are not in the area of optimisation may tend to pick off-the-shelf algorithms to use for their applied problems. This thus does not include some recently-developed advanced/composite algorithms.

## ETHICS STATEMENT

This work complies with the ICLR Code of Ethics. No human subjects, personal data, or sensitive information are involved, and no risks of harm are anticipated.

## REPRODUCIBILITY STATEMENT

Implementation details and experimental settings are provided in Appendix A.1. The data and code are available at an anonymised repository for reproducibility: `https://anonymous.4open.science/r/W2W-E811`.

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

# Appendix to:

# When to Use Which? An Investigation of Search Methods on Expensive Black-box Optimisation Problems

## A  EXPERIMENTAL SETTINGS

### A.1  METHOD DETAILS

- **Vanilla Bayesian Optimisation (vanilla BO)** (Kushner, 1964) is a classical algorithm for expensive BBO problems. Vanilla BO consists of two key components: a probabilistic surrogate model and an acquisition function. In each iteration, a surrogate model is fitted to all observations of the target function made so far. Then, an acquisition function is optimised to determine the most promising solution.

- **Trust Region Bayesian Optimisation (TuRBO)** (Eriksson et al., 2019) is an advanced variant of BO that addresses the limitations of classical BO in high-dimensional spaces. It maintains multiple local models within "trust regions" around the best solutions, dynamically adjusting these regions based on optimisation performance. TuRBO has been deemed as one of the most promising high-dimensional BO algorithms (Santoni et al., 2024; Rashidi et al., 2024).

- **Covariance Matrix Adaptation Evolution Strategy (CMA-ES)** (Hansen & Ostermeier, 2001) is a representative of evolutionary algorithms for BBO. It adapts the covariance matrix of a multivariate normal distribution for sampling candidate solutions based on previously evaluated samples. CMA-ES is widely applied in a variety of fields (Uchida et al., 2024; Ghahremani et al., 2020; Maki et al., 2020) and has been found to be effective in expensive BBO (Loshchilov, 2013; Andersson et al., 2015).

- **Genetic Algorithm (GA)** (Holland, 1992) is one of the most canonical evolutionary algorithms. It uses selection, crossover, and mutation to simulate the process of natural evolution, aiming to find an optimal or near-optimal solution.

- **Differential Evolution (DE)** (Storn & Price, 1997) is another important type of evolutionary algorithms. It optimises a problem by maintaining a population of candidate solutions and creating new solutions by using unit vectors to move across the domain space. DE has been found to be particularly suitable for some real-valued BBO problems (Goudos et al., 2011).

- **Nelder Mead (NM)** (Nelder & Mead, 1965) is a widely used local search heuristic for solving BBO problems. It involves constructing a simplex, e.g., a triangle in two dimensions or a tetrahedron in three dimensions, and iteratively adjusting its vertices to approach the optimum.

- **Random Search (RS)** (Karnopp, 1963) is arguably the simplest search method. RS has been found to perform well in some applied fields (e.g., automated machine learning (Bergstra & Bengio, 2012)). Here, we use RS as a baseline.

All the algorithms use their commonly used/recommended settings. In GA, Simulated Binary Crossover (SBX) and Polynomial Mutation (PM) are used. We set the crossover and mutation rates to 1.0 and $1/D$, respectively, and the distribution indexes for crossover and mutation are set to 2 and 20, respectively (Deb et al., 1996; Deb & Deb, 2014). For DE, a well-known classical DE variant (DE/rand/1/bin) is used (Qin & Li, 2013). Following the common practice (Storn & Price, 1997), we set the crossover rate and mutation scale factor to 0.9 and 0.5, respectively. Regarding CMA-ES, we set the initial step size to 0.2 (20% of the search domain) (Hansen, 2019; Vermetten et al., 2022)). Regarding the population size for the population-based algorithms, we set it to 100 for GA and

DE (Roeva et al., 2015; Piotrowski, 2017) and $4 + \lfloor 3\log(D) \rfloor$ for CMA-ES (Varelas et al., 2018). In vanilla BO, we use the commonly used expected improvement acquisition function (Mockus et al., 1978; Jones et al., 1998) together with the RBF from BoTorch's default implementation. For TuRBO, we follow the authors' recommendations and employ Thompson Sampling and a Matérn 5/2 kernel (Eriksson et al., 2019), and we adopt a batch size of 5 (suggested by Santoni et al. (2024)). In both BO methods, kernel lengthscales are learned by maximising the marginal log-likelihood.

The code of RS, NM, GA, DE, and CMA-ES is taken from the Python modules numpy, scipy (Virtanen et al., 2020), pymoo (Blank & Deb, 2020), DEAP (Fortin et al., 2012), and pycma (Hansen et al., 2019). The code of vanilla BO and TuRBO is from the Python module BoTorch (Balandat et al., 2020). The computational studies were performed on a Red Hat Enterprise Linux 8.8 system, operating on a 64-bit x86 CPU architecture. The computing cluster utilised Intel Xeon Platinum 8360Y processors running at 2.40 GHz.

## A.2 Problem Details

Following the suggestion in Hansen et al. (2009), we categorise the 24 BBOB functions into 5 groups based on their characteristics. They are

- Functions with separate variables (f1-f5);
    - f1 Sphere: A simple convex quadratic, unimodal and perfectly symmetric.
    - f2 Ellipsoidal: A separable quadratic with high conditioning and smooth irregularities.
    - f3 Rastrigin: A highly multimodal separable function with about $10^D$ regularly spaced local optima.
    - f4 Büche-Rastrigin: A deceptive, asymmetric variant of Rastrigin with skewed optima placement.
    - f5 Linear Slope: A linear, boundary-optimal function testing search beyond the initial convex hull.
- Functions with low or moderate conditioning (mild slope) and few or no local optima (f6-f9);
    - f6 Attractive Sector: An asymmetric cone-shaped unimodal landscape with optimum at the tip.
    - f7 Step Ellipsoidal: A plateau-rich ellipsoidal function where gradients vanish almost everywhere.
    - f8 Rosenbrock (original): The classic banana-shaped valley requiring long curved path following.
    - f9 Rosenbrock (rotated): A non-separable rotated version of Rosenbrock, eliminating coordinate structure.
- Functions with high conditioning (steep slope) and few or no local optima (f10-f14);
    - f10 Ellipsoidal (rotated): A non-separable quadratic with strong ill-conditioning ($10^6$).
    - f11 Discus: A quadratic where one direction is far more sensitive than all others.
    - f12 Bent Cigar: A narrow ridge with strong conditioning and asymmetric deformation.
    - f13 Sharp Ridge: A ridge function with non-differentiable shape requiring axis-aligned search.
    - f14 Different Powers: A unimodal function with variables of increasing sensitivity near the optimum.
- Multi-modal functions with adequate global structure (f15-f19);
    - f15 Rastrigin (rotated): A non-separable variant of Rastrigin with alleviated symmetry.
    - f16 Weierstrass: A rugged, fractal-like function with non-unique global optima.
    - f17 Schaffers F7: A multimodal function with varying frequency and amplitude in modulation.
    - f18 Schaffers F7 (ill-conditioned): A moderately ill-conditioned version of f17.
    - f19 Composite Griewank–Rosenbrock (F8F2): A Rosenbrock-like valley embedded in a highly multimodal landscape.

- Multi-modal functions with weak global structure (f20-f24).
    - f20 Schwefel: A deceptive function with prominent local minima near the search-space corners.
    - f21 Gallagher 101-me Peaks: A random landscape of 101 Gaussian peaks without global structure.
    - f22 Gallagher 21-hi Peaks: A harder variant of f21 with 21 peaks and stronger conditioning.
    - f23 Katsuura: A highly rugged and repetitive function with infinitely many optima.
    - f24 Lunacek bi-Rastrigin: A deceptive double-funnel function superimposed with Rastrigin structure.

Their mathematical formulations are provided in Hansen et al. (2009).

## B  DETAILED EXPERIMENTAL RESULTS

### B.1  TABULATED RESULTS

In this section, we detail the statistical results (mean and standard deviation) regarding the regret value for the seven algorithms on 24 BBOB functions with a moderate number of dimensions (10D) under the four budgets (i.e., 200, 800, 2,000, 8,000), shown in Tables 3-6.

Table 3: The statistical results (mean and standard deviation) of the regret (i.e., the difference between the true optimum and the best function value obtained) were computed for the seven methods across the 24 BBOB functions, under a budget of 200 evaluations over 30 independent runs. Methods that are statistically the best are highlighted in **bold**.

| Method | f1 (10D) | | f2 (10D) | | f3 (10D) | | f4 (10D) | | f5 (10D) | | f6 (10D) | |
|---|---|---|---|---|---|---|---|---|---|---|---|---|
| | Mean | Std | Mean | Std | Mean | Std | Mean | Std | Mean | Std | Mean | Std |
| RS | 2.76e+1 | 8.4e+0 | 3.89e+5 | 1.6e+5 | 1.96e+2 | 3.6e+1 | 2.65e+2 | 5.0e+1 | 8.92e+1 | 1.5e+1 | 1.78e+4 | 1.5e+4 |
| NM | 2.39e+1 | 2.1e+1 | 5.99e+5 | 8.4e+5 | 4.40e+2 | 1.7e+2 | 4.50e+2 | 1.4e+2 | 5.00e+1 | 3.1e+1 | 2.18e+4 | 7.0e+4 |
| GA | 1.84e+1 | 4.7e+0 | 1.86e+5 | 1.0e+5 | 1.56e+2 | 3.1e+1 | 2.21e+2 | 3.5e+1 | 7.37e+1 | 1.0e+1 | 2.64e+3 | 3.8e+3 |
| DE | 2.80e+1 | 7.2e+0 | 3.22e+5 | 1.4e+5 | 2.04e+2 | 3.2e+1 | 2.65e+2 | 4.4e+1 | 7.65e+1 | 2.0e+1 | 2.12e+4 | 2.3e+4 |
| CMA-ES | 5.98e+0 | 3.5e+0 | 9.32e+4 | 6.0e+4 | 1.11e+2 | 2.7e+1 | 1.59e+2 | 3.5e+1 | 2.26e+1 | 7.7e+0 | 1.14e+2 | 3.7e+1 |
| Vanilla BO | **1.74e−3** | **3.4e−4** | **1.04e+3** | **2.8e+2** | 8.47e+1 | 8.9e+0 | 1.39e+2 | 9.4e+0 | **1.33e−1** | **5.9e−2** | 6.90e+1 | 4.1e−1 |
| TuRBO | 2.86e−2 | 5.9e−3 | 3.04e+3 | 1.4e+3 | **5.11e+1** | **1.3e+1** | **1.33e+2** | **1.2e+1** | 1.17e+0 | 3.7e−1 | **2.23e+1** | **1.7e+1** |

| Method | f7 (10D) | | f8 (10D) | | f9 (10D) | | f10 (10D) | | f11 (10D) | | f12 (10D) | |
|---|---|---|---|---|---|---|---|---|---|---|---|---|
| | Mean | Std | Mean | Std | Mean | Std | Mean | Std | Mean | Std | Mean | Std |
| RS | 1.62e+2 | 5.0e+1 | 1.51e+4 | 7.6e+3 | 1.23e+4 | 5.0e+3 | 4.34e+5 | 2.8e+5 | 7.18e+2 | 8.0e+2 | 3.12e+7 | 8.7e+6 |
| NM | 1.26e+3 | 8.7e+2 | 1.22e+4 | 2.0e+4 | 1.27e+4 | 1.8e+4 | 4.71e+5 | 8.7e+5 | 1.10e+2 | 5.0e+1 | 6.77e+7 | 6.0e+7 |
| GA | 1.06e+2 | 3.3e+1 | 8.13e+3 | 3.4e+3 | 5.78e+3 | 3.0e+3 | 2.06e+5 | 1.2e+5 | 2.08e+2 | 1.7e+2 | 2.45e+7 | 7.7e+6 |
| DE | 1.48e+2 | 5.3e+1 | 1.81e+4 | 7.3e+3 | 1.07e+4 | 5.2e+3 | 3.87e+5 | 2.0e+5 | 8.84e+2 | 1.2e+3 | 3.28e+7 | 8.3e+6 |
| CMA-ES | 4.84e+1 | 2.2e+1 | 1.81e+3 | 2.4e+3 | 1.44e+3 | 1.3e+3 | 1.33e+5 | 1.0e+5 | 2.99e+2 | 3.3e+2 | 6.39e+6 | 5.3e+6 |
| Vanilla BO | **1.04e+1** | **1.4e+0** | **8.60e+1** | **4.1e+1** | 2.36e+2 | 9.7e+1 | 2.95e+4 | 6.9e+3 | **8.76e+1** | **2.7e+1** | 1.83e+7 | 6.3e+6 |
| TuRBO | 1.37e+1 | 1.4e+1 | 1.12e+2 | 6.0e+1 | **7.76e+1** | **6.0e+1** | **9.22e+3** | **2.7e+3** | 1.21e+2 | 3.3e+0 | **2.54e+5** | **4.2e+5** |

| Method | f13 (10D) | | f14 (10D) | | f15 (10D) | | f16 (10D) | | f17 (10D) | | f18 (10D) | |
|---|---|---|---|---|---|---|---|---|---|---|---|---|
| | Mean | Std | Mean | Std | Mean | Std | Mean | Std | Mean | Std | Mean | Std |
| RS | 1.00e+3 | 1.5e+2 | 1.25e+1 | 3.9e+0 | 2.25e+2 | 3.8e+1 | 2.08e+1 | 5.3e+0 | 8.89e+0 | 2.2e+0 | 3.39e+1 | 9.5e+0 |
| NM | 7.12e+2 | 3.3e+2 | 2.36e+1 | 2.0e+1 | 5.07e+2 | 2.0e+2 | 3.13e+1 | 1.8e+1 | 5.06e+1 | 5.3e+1 | 3.46e+2 | 3.9e+2 |
| GA | 7.65e+2 | 1.4e+2 | 9.46e+0 | 2.5e+0 | 1.75e+2 | 2.5e+1 | 1.91e+1 | 5.1e+0 | 7.21e+0 | 1.4e+0 | 2.72e+1 | 6.5e+0 |
| DE | 1.03e+3 | 1.7e+2 | 1.23e+1 | 3.5e+0 | 2.36e+2 | 3.8e+1 | 2.24e+1 | 4.8e+0 | 1.00e+1 | 2.1e+0 | 3.96e+1 | 8.6e+0 |
| CMA-ES | 5.11e+2 | 1.4e+2 | 5.03e+0 | 2.8e+0 | 1.15e+2 | 3.2e+1 | 2.19e+1 | 5.0e+0 | 4.67e+0 | 1.1e+0 | 1.76e+1 | 4.4e+0 |
| Vanilla BO | 4.94e+1 | 4.8e+0 | 3.89e−1 | 1.7e−1 | 1.16e+2 | 2.1e+1 | **5.37e+0** | **3.3e+0** | **2.44e+0** | **7.8e−1** | 1.18e+1 | 1.2e+0 |
| TuRBO | **3.21e+1** | **7.1e+0** | **3.05e−1** | **2.7e−1** | 8.05e+1 | 1.9e+1 | 7.73e+0 | 2.9e+0 | **2.20e+0** | **1.0e+0** | **1.03e+1** | **1.4e+0** |

| Method | f19 (10D) | | f20 (10D) | | f21 (10D) | | f22 (10D) | | f23 (10D) | | f24 (10D) | |
|---|---|---|---|---|---|---|---|---|---|---|---|---|
| | Mean | Std | Mean | Std | Mean | Std | Mean | Std | Mean | Std | Mean | Std |
| RS | 1.24e+1 | 2.3e+0 | 5.40e+3 | 2.2e+3 | 4.83e+1 | 1.1e+1 | 5.81e+1 | 1.4e+1 | 3.64e+0 | 7.9e−1 | 1.59e+2 | 2.1e+1 |
| NM | 2.84e+1 | 1.7e+1 | 1.95e+4 | 2.9e+4 | 3.46e+1 | 2.1e+1 | 4.99e+1 | 2.8e+1 | **1.44e+0** | **7.4e−1** | 2.59e+2 | 6.3e+1 |
| GA | 9.01e+0 | 2.1e+0 | 2.84e+3 | 2.1e+3 | 4.01e+1 | 1.1e+1 | 4.90e+1 | 1.4e+1 | 3.09e+0 | 6.8e−1 | 1.37e+2 | 1.6e+1 |
| DE | 1.20e+1 | 2.5e+0 | 6.67e+3 | 4.1e+3 | 4.74e+1 | 1.1e+1 | 5.80e+1 | 1.2e+1 | 3.55e+0 | 7.9e−1 | 1.75e+2 | 2.0e+1 |
| CMA-ES | **6.70e+0** | **1.8e+0** | 4.70e+2 | 9.6e+2 | 2.04e+1 | 1.3e+1 | 2.64e+1 | 1.6e+1 | 3.31e+0 | 8.3e−1 | 1.05e+2 | 2.2e+1 |
| Vanilla BO | 6.37e+0 | 1.5e+0 | **2.75e+0** | **2.6e−1** | 2.22e+1 | 9.7e+0 | 9.29e+0 | 6.2e+0 | 2.95e+0 | 4.7e−1 | 7.95e+1 | 6.6e+0 |
| TuRBO | 6.54e+0 | 2.3e−1 | 2.82e+0 | 2.3e−1 | **6.51e+0** | **2.5e+0** | **4.11e+0** | **3.2e+0** | 3.07e+0 | 4.3e−1 | **7.39e+1** | **8.8e+0** |

Table 4: The statistical results (mean and standard deviation) of the regret (i.e., the difference between the true optimum and the best function value obtained) were computed for the seven methods across the 24 BBOB functions, under a budget of 800 evaluations over 30 independent runs. Methods that are statistically the best are highlighted in **bold**.

| Method | f1 (10D) | | f2 (10D) | | f3 (10D) | | f4 (10D) | | f5 (10D) | | f6 (10D) | |
|---|---|---|---|---|---|---|---|---|---|---|---|---|
| | Mean | Std | Mean | Std | Mean | Std | Mean | Std | Mean | Std | Mean | Std |
| RS | 1.99e+1 | 5.4e+0 | 2.28e+5 | 1.0e+5 | 1.64e+2 | 2.2e+1 | 2.19e+2 | 3.3e+1 | 7.31e+1 | 9.3e+0 | 3.91e+3 | 4.3e+3 |
| NM | 7.96e+0 | 1.0e+1 | 1.15e+5 | 2.6e+5 | 4.22e+2 | 1.7e+2 | 4.32e+2 | 1.4e+2 | **1.37e+1** | **2.5e+1** | 1.01e+3 | 3.8e+3 |
| GA | 3.38e+0 | 1.0e+0 | 2.12e+4 | 1.6e+4 | 7.11e+1 | 1.2e+1 | 9.77e+1 | 1.7e+1 | 2.83e+1 | 4.4e+0 | 7.35e+1 | 2.4e+1 |
| DE | 1.31e+1 | 3.1e+0 | 8.85e+4 | 4.0e+4 | 1.45e+2 | 2.0e+1 | 2.03e+2 | 3.3e+1 | 1.98e+1 | 1.0e+1 | 2.38e+2 | 1.0e+2 |
| CMA-ES | **8.91e−4** | **7.8e−4** | 4.29e+3 | 6.5e+3 | 4.70e+1 | 1.3e+1 | **5.33e+1** | **1.5e+1** | **2.31e−2** | **2.5e−2** | **8.21e+0** | **5.0e+0** |
| Vanilla BO | 1.74e−3 | 3.4e−4 | **1.65e+2** | **1.3e+1** | 5.19e+1 | 1.1e+0 | 1.04e+2 | 1.0e+1 | 1.33e−1 | 5.9e−2 | 5.61e+1 | 4.3e−1 |
| TuRBO | 1.30e−2 | 1.3e−3 | 2.94e+3 | 1.1e+3 | **4.10e+1** | **4.5e+0** | 1.23e+2 | 1.9e+1 | 4.34e−1 | 5.7e−2 | 1.39e+1 | 2.2e+0 |

| Method | f7 (10D) | | f8 (10D) | | f9 (10D) | | f10 (10D) | | f11 (10D) | | f12 (10D) | |
|---|---|---|---|---|---|---|---|---|---|---|---|---|
| | Mean | Std | Mean | Std | Mean | Std | Mean | Std | Mean | Std | Mean | Std |
| RS | 9.17e+1 | 2.5e+1 | 8.25e+3 | 2.7e+3 | 6.58e+3 | 2.8e+3 | 2.15e+5 | 1.1e+5 | 1.27e+2 | 6.1e+1 | 1.98e+7 | 6.6e+6 |
| NM | 5.39e+2 | 4.3e+2 | 3.07e+3 | 4.9e+3 | 2.65e+3 | 4.7e+3 | 5.33e+4 | 1.1e+5 | **7.28e+1** | **4.1e+1** | 2.43e+7 | 2.7e+7 |
| GA | 2.65e+1 | 7.4e+0 | 7.87e+2 | 4.4e+2 | 5.18e+2 | 2.7e+2 | 5.35e+4 | 2.1e+4 | **8.33e+1** | **4.1e+1** | 3.52e+6 | 1.3e+6 |
| DE | 7.86e+1 | 2.6e+1 | 5.36e+3 | 2.8e+3 | 3.20e+3 | 1.8e+3 | 1.53e+5 | 6.3e+4 | 1.57e+2 | 6.8e+1 | 1.78e+7 | 6.9e+6 |
| CMA-ES | 4.02e+0 | 2.6e+0 | **1.22e+1** | **1.3e+1** | **2.55e+1** | **3.5e+1** | 1.34e+4 | 1.2e+4 | 1.06e+2 | 4.3e+1 | **2.22e+3** | **3.9e+3** |
| Vanilla BO | **3.88e+0** | **6.3e−1** | 1.76e+1 | 1.1e+1 | 4.36e+1 | 1.1e+1 | 4.58e+3 | 9.2e+2 | **7.70e+1** | **8.7e+0** | 1.08e+7 | 7.0e+6 |
| TuRBO | 5.43e+0 | 1.2e+0 | 5.01e+1 | 1.3e+1 | 5.31e+1 | 2.0e+1 | **3.00e+3** | **1.6e+3** | 9.22e+1 | 1.4e+1 | 1.06e+5 | 1.2e+5 |

| Method | f13 (10D) | | f14 (10D) | | f15 (10D) | | f16 (10D) | | f17 (10D) | | f18 (10D) | |
|---|---|---|---|---|---|---|---|---|---|---|---|---|
| | Mean | Std | Mean | Std | Mean | Std | Mean | Std | Mean | Std | Mean | Std |
| RS | 7.87e+2 | 1.1e+2 | 8.44e+0 | 2.4e+0 | 1.74e+2 | 2.1e+1 | 1.65e+1 | 3.7e+0 | 7.16e+0 | 1.4e+0 | 2.68e+1 | 5.5e+0 |
| NM | 2.62e+2 | 2.7e+2 | 9.97e+0 | 1.2e+1 | 4.88e+2 | 1.9e+2 | 3.08e+1 | 1.8e+1 | 4.37e+1 | 4.3e+1 | 2.88e+2 | 3.2e+2 |
| GA | 3.39e+2 | 5.8e+1 | 2.58e+0 | 9.8e−1 | 8.80e+1 | 1.1e+1 | 1.43e+1 | 3.6e+0 | 2.68e+0 | 6.4e−1 | 1.08e+1 | 2.1e+0 |
| DE | 7.50e+2 | 1.1e+2 | 7.47e+0 | 2.0e+0 | 1.57e+2 | 2.8e+1 | 1.93e+1 | 4.4e+0 | 6.96e+0 | 1.4e+0 | 2.58e+1 | 5.7e+0 |
| CMA-ES | 2.44e+1 | 2.1e+1 | **2.75e−2** | **2.1e−2** | **5.70e+1** | **1.0e+1** | 1.37e+1 | 6.5e+0 | **6.38e−1** | **5.2e−1** | **2.38e+0** | **1.4e+0** |
| Vanilla BO | 4.16e+1 | 4.5e+0 | 7.97e−2 | 2.6e−2 | 6.38e+1 | 8.2e+0 | **1.64e+0** | **1.2e+0** | 1.42e+0 | 3.1e−1 | 4.39e+0 | 8.1e−1 |
| TuRBO | **1.99e+1** | **2.1e+0** | 1.45e−1 | 3.4e−2 | 6.77e+1 | 5.7e+0 | 3.02e+0 | 4.3e−1 | 1.26e+0 | 4.6e−1 | 3.89e+0 | 5.5e−1 |

| Method | f19 (10D) | | f20 (10D) | | f21 (10D) | | f22 (10D) | | f23 (10D) | | f24 (10D) | |
|---|---|---|---|---|---|---|---|---|---|---|---|---|
| | Mean | Std | Mean | Std | Mean | Std | Mean | Std | Mean | Std | Mean | Std |
| RS | 1.01e+1 | 1.4e+0 | 2.97e+3 | 1.5e+3 | 3.29e+1 | 1.0e+1 | 4.23e+1 | 1.2e+1 | 2.69e+0 | 6.7e−1 | 1.36e+2 | 1.6e+1 |
| NM | 2.62e+1 | 1.7e+1 | 1.02e+4 | 2.1e+4 | 1.82e+1 | 1.6e+1 | 3.20e+1 | 2.7e+1 | **1.17e+0** | **7.4e−1** | 2.49e+2 | 6.1e+1 |
| GA | 5.16e+0 | 6.8e−1 | 2.49e+1 | 8.3e+1 | 1.46e+1 | 5.6e+0 | 1.12e+1 | 4.7e+0 | 2.50e+0 | 4.9e−1 | 8.08e+1 | 1.4e+1 |
| DE | 7.65e+0 | 1.7e+0 | 1.16e+3 | 1.1e+3 | 3.57e+1 | 1.2e+1 | 3.85e+1 | 1.6e+1 | 3.17e+0 | 6.6e−1 | 1.29e+2 | 1.9e+1 |
| CMA-ES | 4.19e+0 | 7.3e−1 | **2.25e+0** | **4.4e−1** | 7.60e+0 | 7.7e+0 | 1.26e+1 | 1.3e+1 | 2.65e+0 | 6.6e−1 | **5.94e+1** | **9.3e+0** |
| Vanilla BO | 4.47e+0 | 4.9e−1 | 2.42e+0 | 1.2e−1 | 4.39e+0 | 3.2e+0 | **2.05e+0** | **9.3e−2** | 2.71e+0 | 2.2e−1 | **6.13e+1** | **8.0e+0** |
| TuRBO | **4.15e+0** | **5.7e−1** | 2.73e+0 | 1.7e−1 | **2.17e+0** | **2.2e−1** | 2.75e+0 | 1.4e+0 | 2.62e+0 | 7.0e−2 | 7.19e+1 | 3.8e+0 |

Table 5: The statistical results (mean and standard deviation) of the regret (i.e., the difference between the true optimum and the best function value obtained) were computed for the seven methods across the 24 BBOB functions, under a budget of 2,000 evaluations over 30 independent runs. Methods that are statistically the best are highlighted in **bold**.

| Method | f1 (10D) | | f2 (10D) | | f3 (10D) | | f4 (10D) | | f5 (10D) | | f6 (10D) | |
|---|---|---|---|---|---|---|---|---|---|---|---|---|
| | Mean | Std | Mean | Std | Mean | Std | Mean | Std | Mean | Std | Mean | Std |
| RS | 1.65e+1 | 3.9e+0 | 1.43e+5 | 6.5e+4 | 1.50e+2 | 2.0e+1 | 1.89e+2 | 2.6e+1 | 6.67e+1 | 1.0e+1 | 1.77e+3 | 2.8e+3 |
| NM | 9.58e−1 | 4.7e+0 | 6.13e+4 | 1.8e+5 | 3.93e+2 | 1.7e+2 | 4.03e+2 | 1.6e+2 | 7.90e+0 | 2.0e+1 | 6.35e+2 | 2.1e+3 |
| GA | 1.61e−1 | 7.5e−2 | 7.36e+2 | 5.6e+2 | 1.97e+1 | 5.1e+0 | 2.80e+1 | 5.8e+0 | 5.40e+0 | 1.1e+0 | 2.68e+1 | 8.3e+0 |
| DE | 3.18e+0 | 1.1e+0 | 1.35e+4 | 5.4e+3 | 8.36e+1 | 1.2e+1 | 1.18e+2 | 1.5e+1 | **0.00e+0** | **0.0e+0** | 1.72e+2 | 6.4e+1 |
| CMA-ES | **9.74e-12** | **1.3e-11** | **3.11e+1** | **3.0e+1** | **1.55e+1** | **7.1e+0** | **2.19e+1** | **9.1e+0** | 4.19e-10 | 6.0e-10 | **1.20e−2** | **1.8e−2** |
| Vanilla BO | 1.74e−3 | 3.4e−4 | 1.65e+2 | 1.3e+1 | 5.19e+1 | 1.1e+0 | 9.78e+1 | 1.3e+1 | 1.33e−1 | 5.9e−2 | 5.61e+1 | 4.3e−1 |
| TuRBO | 1.30e−2 | 1.3e−3 | 2.94e+3 | 1.1e+3 | 4.10e+1 | 4.5e+0 | 1.23e+2 | 1.9e+1 | 4.34e−1 | 5.7e−2 | 1.37e+1 | 1.7e+0 |

| Method | f7 (10D) | | f8 (10D) | | f9 (10D) | | f10 (10D) | | f11 (10D) | | f12 (10D) | |
|---|---|---|---|---|---|---|---|---|---|---|---|---|
| | Mean | Std | Mean | Std | Mean | Std | Mean | Std | Mean | Std | Mean | Std |
| RS | 7.08e+1 | 1.9e+1 | 6.03e+3 | 2.4e+3 | 4.36e+3 | 2.2e+3 | 1.45e+5 | 5.1e+4 | 8.07e+1 | 2.3e+1 | 1.67e+7 | 5.5e+6 |
| NM | 3.68e+2 | 3.7e+2 | 7.59e+2 | 2.0e+3 | 5.79e+2 | 1.4e+3 | 1.89e+4 | 6.0e+4 | **5.57e+1** | **3.6e+1** | 8.10e+6 | 1.6e+7 |
| GA | 6.47e+0 | 3.0e+0 | 6.33e+1 | 3.4e+1 | 7.41e+1 | 5.4e+1 | 1.67e+4 | 9.6e+3 | **5.55e+1** | **2.5e+1** | 1.85e+5 | 1.3e+5 |
| DE | 2.84e+1 | 9.0e+0 | 6.54e+2 | 3.7e+2 | 5.47e+2 | 1.9e+2 | 4.84e+4 | 1.7e+4 | 1.03e+2 | 2.9e+1 | 4.18e+6 | 1.8e+6 |
| CMA-ES | **1.94e+0** | **1.9e+0** | **5.08e+0** | **2.0e+0** | **1.05e+1** | **2.3e+1** | **4.50e+2** | **4.8e+2** | 6.52e+1 | 3.9e+1 | **1.11e+1** | **1.9e+1** |
| Vanilla BO | 3.88e+0 | 6.3e−1 | 1.57e+1 | 9.3e+0 | 2.57e+1 | 5.0e+0 | 3.65e+3 | 9.3e+2 | 7.70e+1 | 8.7e+0 | 1.07e+7 | 7.3e+6 |
| TuRBO | 5.43e+0 | 1.2e+0 | 3.91e+1 | 1.2e+1 | 4.71e+1 | 1.5e+1 | 3.00e+3 | 1.6e+3 | 9.22e+1 | 1.4e+1 | 8.65e+4 | 2.7e+4 |

| Method | f13 (10D) | | f14 (10D) | | f15 (10D) | | f16 (10D) | | f17 (10D) | | f18 (10D) | |
|---|---|---|---|---|---|---|---|---|---|---|---|---|
| | Mean | Std | Mean | Std | Mean | Std | Mean | Std | Mean | Std | Mean | Std |
| RS | 7.00e+2 | 9.2e+1 | 7.13e+0 | 1.4e+0 | 1.60e+2 | 2.5e+1 | 1.43e+1 | 3.2e+0 | 6.39e+0 | 1.0e+0 | 2.28e+1 | 3.5e+0 |
| NM | 9.06e+1 | 1.3e+2 | 5.92e+0 | 8.2e+0 | 4.32e+2 | 1.9e+2 | 2.31e+1 | 1.2e+1 | 4.14e+1 | 4.3e+1 | 2.74e+2 | 2.8e+2 |
| GA | 8.05e+1 | 1.8e+1 | 2.54e−1 | 2.0e−1 | 5.05e+1 | 7.6e+0 | 8.65e+0 | 3.8e+0 | 8.30e−1 | 2.6e−1 | 3.33e+0 | 1.3e+0 |
| DE | 3.70e+2 | 6.5e+1 | 2.70e+0 | 6.6e−1 | 9.15e+1 | 1.4e+1 | 1.70e+1 | 4.0e+0 | 3.17e+0 | 6.8e−1 | 1.18e+1 | 2.6e+0 |
| CMA-ES | **4.77e+0** | **5.6e+0** | **6.67e−5** | **4.9e−5** | **1.81e+1** | **8.0e+0** | **6.68e+0** | **6.7e+0** | **6.80e−2** | **1.0e−1** | **3.21e−1** | **3.9e−1** |
| Vanilla BO | 4.16e+1 | 4.5e+0 | 7.97e−2 | 2.6e−2 | 6.38e+1 | 8.2e+0 | **1.64e+0** | **1.2e+0** | 1.30e+0 | 2.7e−1 | 4.21e+0 | 8.7e−1 |
| TuRBO | 1.95e+1 | 1.2e+0 | 7.72e−2 | 5.0e−2 | 5.87e+1 | 3.1e+0 | 3.02e+0 | 4.3e−1 | 1.26e+0 | 4.6e−1 | 3.81e+0 | 3.2e−1 |

| Method | f19 (10D) | | f20 (10D) | | f21 (10D) | | f22 (10D) | | f23 (10D) | | f24 (10D) | |
|---|---|---|---|---|---|---|---|---|---|---|---|---|
| | Mean | Std | Mean | Std | Mean | Std | Mean | Std | Mean | Std | Mean | Std |
| RS | 8.96e+0 | 1.4e+0 | 1.99e+3 | 1.2e+3 | 2.54e+1 | 7.6e+0 | 3.18e+1 | 1.1e+1 | 2.08e+0 | 4.5e−1 | 1.26e+2 | 1.4e+1 |
| NM | 1.97e+1 | 1.3e+1 | 5.60e+3 | 1.6e+4 | 1.44e+1 | 1.5e+1 | 2.50e+1 | 2.4e+1 | **6.79e−1** | **3.0e−1** | 2.29e+2 | 6.1e+1 |
| GA | 3.59e+0 | 7.4e−1 | **1.82e+0** | **3.6e−1** | 4.41e+0 | 3.0e+0 | 2.24e+0 | 1.1e+0 | 2.06e+0 | 4.8e−1 | 5.88e+1 | 5.0e+0 |
| DE | 5.73e+0 | 8.8e−1 | 3.60e+0 | 4.9e−1 | 2.11e+1 | 7.2e+0 | 1.92e+1 | 9.2e+0 | 2.79e+0 | 5.6e−1 | 8.61e+1 | 1.2e+1 |
| CMA-ES | **3.19e+0** | **6.5e−1** | **1.68e+0** | **3.5e−1** | 7.58e+0 | 7.7e+0 | **1.26e+1** | **1.3e+1** | 2.25e+0 | 4.2e−1 | **4.92e+1** | **8.5e+0** |
| Vanilla BO | 4.33e+0 | 4.4e−1 | 2.34e+0 | 1.6e−1 | **2.01e+0** | **2.2e−1** | 2.03e+0 | 3.6e−2 | 2.57e+0 | 3.6e−1 | 5.77e+1 | 9.0e+0 |
| TuRBO | 4.15e+0 | 5.7e−1 | 2.73e+0 | 1.7e−1 | **2.03e+0** | **4.7e−1** | 2.15e+0 | 2.9e−1 | 2.62e+0 | 7.0e−2 | 7.15e+1 | 4.1e+0 |

Table 6: The statistical results (mean and standard deviation) of the regret (i.e., the difference between the true optimum and the best function value obtained) were computed for the seven methods across the 24 BBOB functions, under a budget of 8,000 evaluations over 30 independent runs. Methods that are statistically the best are highlighted in **bold**.

| Method | f1 (10D) | | f2 (10D) | | f3 (10D) | | f4 (10D) | | f5 (10D) | | f6 (10D) | |
|---|---|---|---|---|---|---|---|---|---|---|---|---|
| | Mean | Std | Mean | Std | Mean | Std | Mean | Std | Mean | Std | Mean | Std |
| RS | 1.19e+1 | 2.7e+0 | 8.11e+4 | 3.8e+4 | 1.23e+2 | 1.7e+1 | 1.67e+2 | 1.7e+1 | 5.66e+1 | 9.1e+0 | 1.66e+2 | 1.3e+2 |
| NM | 1.02e-10 | 5.1e-11 | 1.20e+4 | 4.4e+4 | 2.18e+2 | 7.2e+1 | 2.81e+2 | 1.0e+2 | **0.00e+0** | **0.0e+0** | 1.02e+2 | 1.3e+2 |
| GA | 2.12e−7 | 3.1e−7 | 4.23e−3 | 7.5e−3 | **2.99e+0** | **1.5e+0** | **4.28e+0** | **1.7e+0** | 1.09e−3 | 6.2e−4 | 3.93e+0 | 2.5e+0 |
| DE | 2.67e−3 | 1.5e−3 | 9.14e+0 | 5.1e+0 | 3.98e+1 | 4.6e+0 | 4.53e+1 | 5.4e+0 | **0.00e+0** | **0.0e+0** | 1.52e+1 | 6.6e+0 |
| CMA-ES | **1.42e-14** | **8.2e-15** | **1.99e-14** | **1.5e-14** | 1.18e+1 | 5.1e+0 | 1.72e+1 | 6.1e+0 | 6.99e-15 | 6.2e-15 | **5.46e-13** | **4.9e-13** |
| Vanilla BO | 1.74e−3 | 3.4e−4 | 1.65e+2 | 1.3e+1 | 5.19e+1 | 1.1e+0 | 9.78e+1 | 1.3e+1 | 1.33e−1 | 5.9e−2 | 5.61e+1 | 4.3e−1 |
| TuRBO | 1.30e−2 | 1.3e−3 | 2.94e+3 | 1.1e+3 | 4.10e+1 | 4.5e+0 | 1.23e+2 | 1.9e+1 | 4.34e−1 | 5.7e−2 | 1.37e+1 | 1.7e+0 |

| Method | f7 (10D) | | f8 (10D) | | f9 (10D) | | f10 (10D) | | f11 (10D) | | f12 (10D) | |
|---|---|---|---|---|---|---|---|---|---|---|---|---|
| | Mean | Std | Mean | Std | Mean | Std | Mean | Std | Mean | Std | Mean | Std |
| RS | 4.93e+1 | 1.4e+1 | 3.13e+3 | 1.3e+3 | 2.69e+3 | 9.8e+2 | 7.92e+4 | 3.4e+4 | 5.93e+1 | 1.4e+1 | 1.21e+7 | 3.2e+6 |
| NM | 1.14e+2 | 8.0e+1 | 2.27e+2 | 1.1e+3 | 1.17e+1 | 2.3e+1 | 8.15e+2 | 7.6e+2 | 2.53e+1 | 1.7e+1 | 2.79e+2 | 7.8e+2 |
| GA | 3.57e+0 | 1.9e+0 | 1.05e+1 | 1.3e+1 | 2.08e+1 | 3.1e+1 | 8.60e+3 | 5.4e+3 | 4.56e+1 | 2.2e+1 | 6.25e+0 | 9.4e+0 |
| DE | 9.41e−1 | 4.1e−1 | 9.85e+0 | 1.2e+0 | 9.77e+0 | 1.4e+0 | 1.61e+3 | 6.3e+2 | 8.00e+0 | 2.8e+0 | 8.61e+3 | 4.7e+3 |
| CMA-ES | **6.17e−1** | **7.2e−1** | **3.99e−1** | **1.2e+0** | **2.58e−1** | **9.5e−1** | **7.26e+0** | **2.3e+1** | **2.09e+0** | **5.9e+0** | **2.02e+0** | **6.7e+0** |
| Vanilla BO | 3.88e+0 | 6.3e−1 | 1.57e+1 | 9.3e+0 | 2.57e+1 | 5.0e+0 | 3.65e+3 | 9.3e+2 | 7.70e+1 | 8.7e+0 | 1.07e+7 | 7.3e+6 |
| TuRBO | 5.43e+0 | 1.2e+0 | 3.91e+1 | 1.2e+1 | 4.71e+1 | 1.5e+1 | 3.00e+3 | 1.6e+3 | 9.22e+1 | 1.4e+1 | 8.65e+4 | 2.7e+4 |

| Method | f13 (10D) | | f14 (10D) | | f15 (10D) | | f16 (10D) | | f17 (10D) | | f18 (10D) | |
|---|---|---|---|---|---|---|---|---|---|---|---|---|
| | Mean | Std | Mean | Std | Mean | Std | Mean | Std | Mean | Std | Mean | Std |
| RS | 6.01e+2 | 7.9e+1 | 5.57e+0 | 1.3e+0 | 1.36e+2 | 1.6e+1 | 1.05e+1 | 2.0e+0 | 4.97e+0 | 7.8e−1 | 1.74e+1 | 3.7e+0 |
| NM | 1.55e+1 | 3.9e+1 | 1.15e−2 | 4.4e−2 | 2.73e+2 | 1.0e+2 | 1.28e+1 | 7.4e+0 | 2.00e+1 | 1.6e+1 | 1.09e+2 | 7.7e+1 |
| GA | 6.76e+0 | 6.4e+0 | 4.55e−3 | 3.3e−3 | 2.11e+1 | 9.0e+0 | 3.63e+0 | 2.0e+0 | 2.11e−1 | 1.8e−1 | 7.32e−1 | 4.9e−1 |
| DE | 2.23e+1 | 5.4e+0 | 1.25e−2 | 4.7e−3 | 4.52e+1 | 6.5e+0 | 1.45e+1 | 2.6e+0 | 2.99e−1 | 6.0e−2 | 1.33e+0 | 3.8e−1 |
| CMA-ES | **2.10e−5** | **7.9e−5** | **8.11e-12** | **6.6e-12** | **1.27e+1** | **6.1e+0** | **1.40e+0** | **2.3e+0** | **3.42e−2** | **7.2e−2** | **1.51e−1** | **2.0e−1** |
| Vanilla BO | 4.16e+1 | 4.5e+0 | 7.97e−2 | 2.6e−2 | 6.38e+1 | 8.2e+0 | 1.64e+0 | 1.2e+0 | 1.30e+0 | 2.7e−1 | 4.21e+0 | 8.7e−1 |
| TuRBO | 1.92e+1 | 1.3e+0 | 7.72e−2 | 5.0e−2 | 5.87e+1 | 3.1e+0 | 3.02e+0 | 4.3e−1 | 1.26e+0 | 4.6e−1 | 3.81e+0 | 3.2e−1 |

| Method | f19 (10D) | | f20 (10D) | | f21 (10D) | | f22 (10D) | | f23 (10D) | | f24 (10D) | |
|---|---|---|---|---|---|---|---|---|---|---|---|---|
| | Mean | Std | Mean | Std | Mean | Std | Mean | Std | Mean | Std | Mean | Std |
| RS | 7.37e+0 | 1.1e+0 | 8.07e+2 | 6.1e+2 | 1.88e+1 | 5.0e+0 | 2.26e+1 | 7.1e+0 | 1.63e+0 | 3.4e−1 | 1.15e+2 | 9.7e+0 |
| NM | 1.13e+1 | 5.0e+0 | 5.79e+2 | 1.3e+3 | **2.75e+0** | **2.7e+0** | 2.10e+0 | 3.0e+0 | **3.81e−1** | **1.3e−1** | 1.60e+2 | 4.1e+1 |
| GA | 1.86e+0 | 5.7e−1 | **7.33e−1** | **2.5e−1** | 2.11e+0 | 1.8e+0 | **1.85e+0** | **4.9e−1** | 1.46e+0 | 3.1e−1 | 2.97e+1 | 7.6e+0 |
| DE | 3.21e+0 | 6.4e−1 | 2.22e+0 | 1.6e−1 | 2.62e+0 | 1.8e+0 | 2.03e+0 | 5.4e−1 | 2.13e+0 | 4.3e−1 | 5.73e+1 | 6.8e+0 |
| CMA-ES | **1.01e+0** | **6.9e−1** | 1.46e+0 | 3.0e−1 | 2.69e+0 | 2.9e+0 | 2.08e+0 | 1.5e+0 | 1.36e+0 | 7.3e−1 | **2.49e+1** | **8.7e+0** |
| Vanilla BO | 4.33e+0 | 4.4e−1 | 2.34e+0 | 1.6e−1 | **2.01e+0** | **2.2e−1** | 2.03e+0 | 3.6e−2 | 2.57e+0 | 3.6e−1 | 5.77e+1 | 9.0e+0 |
| TuRBO | 4.15e+0 | 5.7e−1 | 2.73e+0 | 1.7e−1 | **2.03e+0** | **4.7e−1** | 2.15e+0 | 2.9e−1 | 2.62e+0 | 7.0e−2 | 7.15e+1 | 4.1e+0 |

## B.2 CONVERGENCE TRAJECTORIES

Figures 6-10 show the convergence trajectories of the seven algorithms throughout the optimisation process on the BBOB f1-f4, f5-f9, f10-f14, f15-f19, and f20-f24, respectively, with the low-dimension (left), medium-dimension (middle), and high-dimension (right).

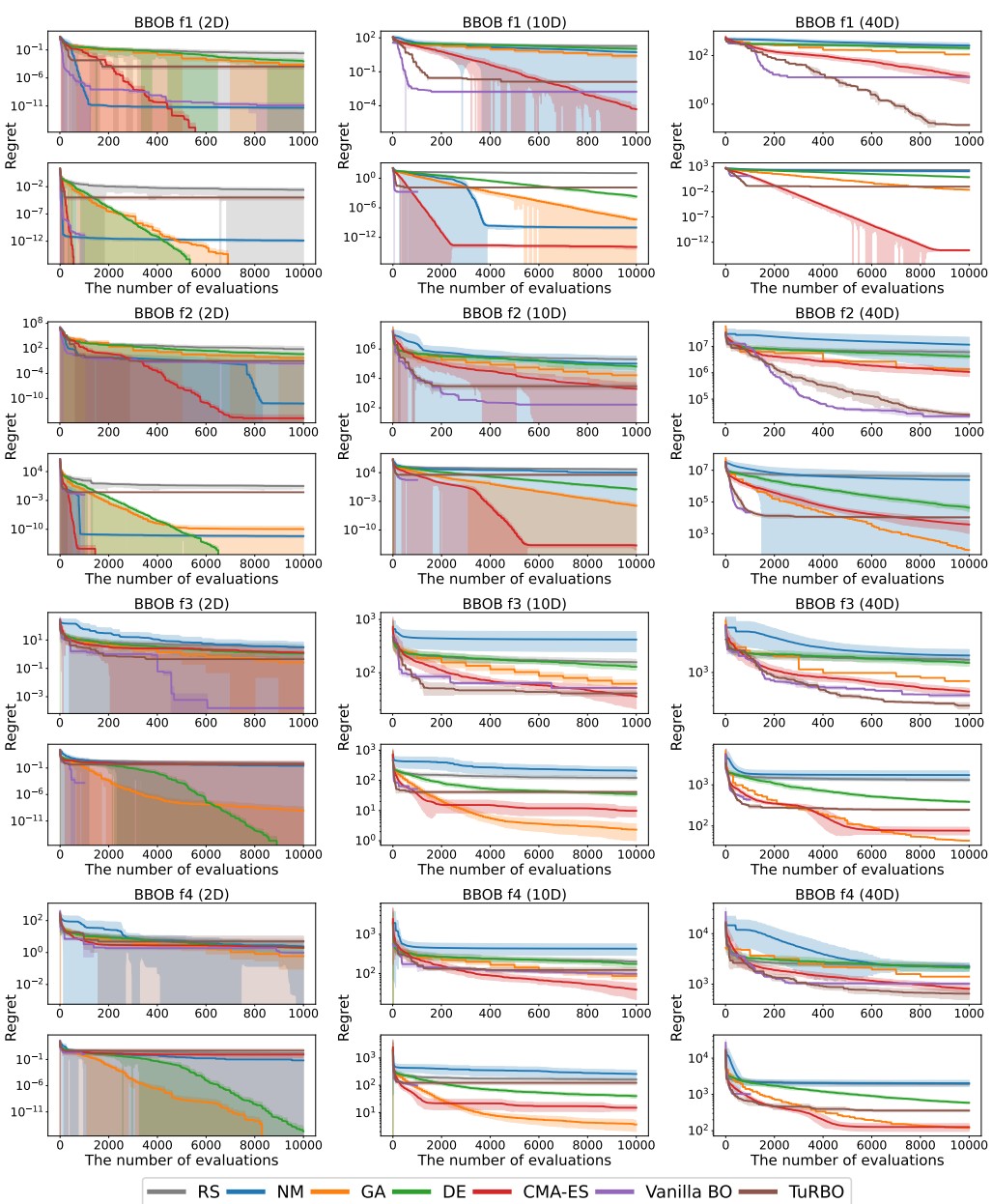

Figure 6: The convergence trajectories of the seven algorithms throughout the optimisation process on the BBOB f1-f4 with the low-dimension (left), medium-dimension (middle), and high-dimension (right). Each coloured line represents the mean regret (the difference between the true optimum and the best function value) obtained over 30 independent runs.

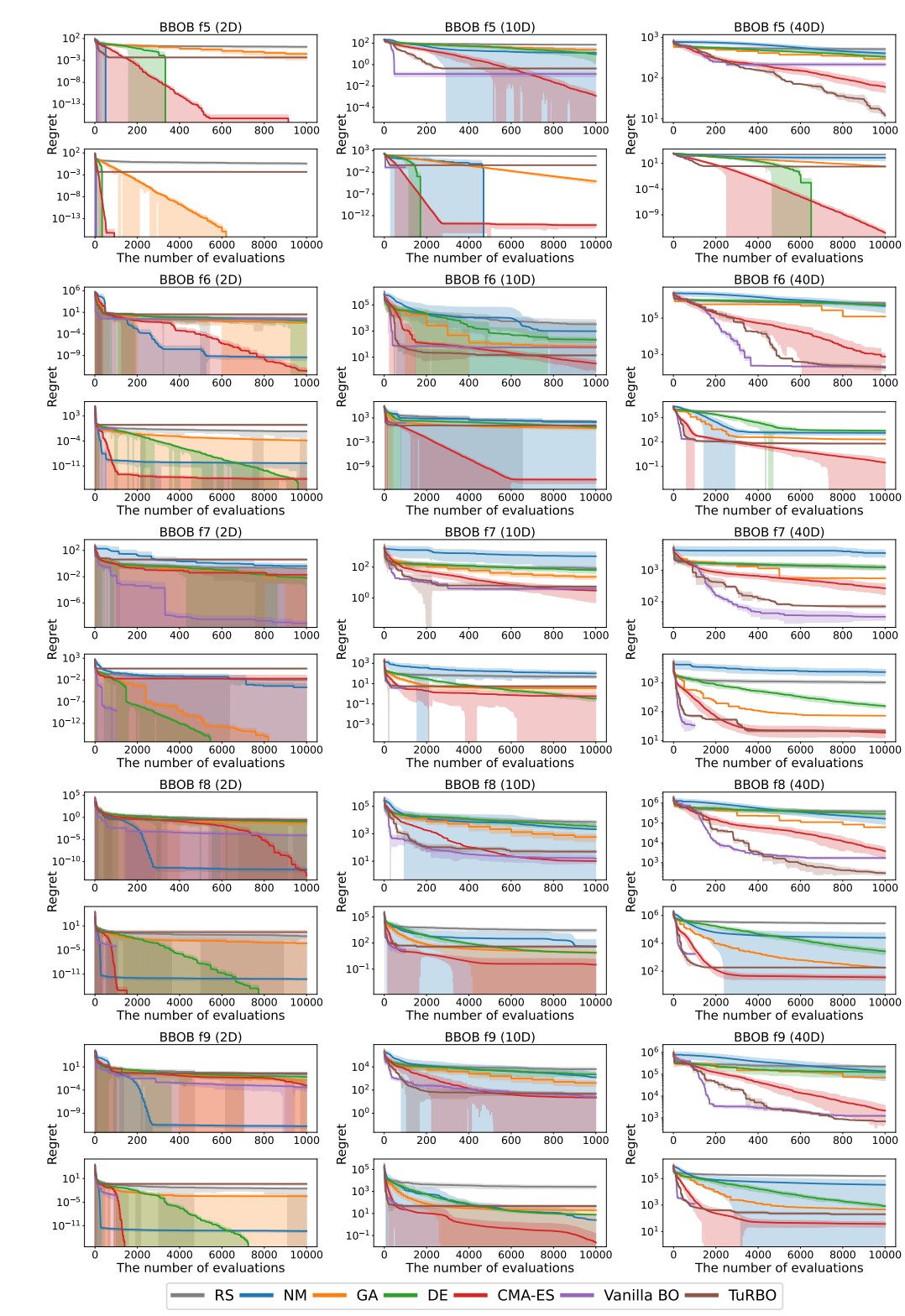

Figure 7: The convergence trajectories of the seven algorithms throughout the optimisation process on the BBOB f5-f9 with the low-dimension (left), medium-dimension (middle), and high-dimension (right). Each coloured line represents the mean regret (the difference between the true optimum and the best function value) obtained over 30 independent runs.

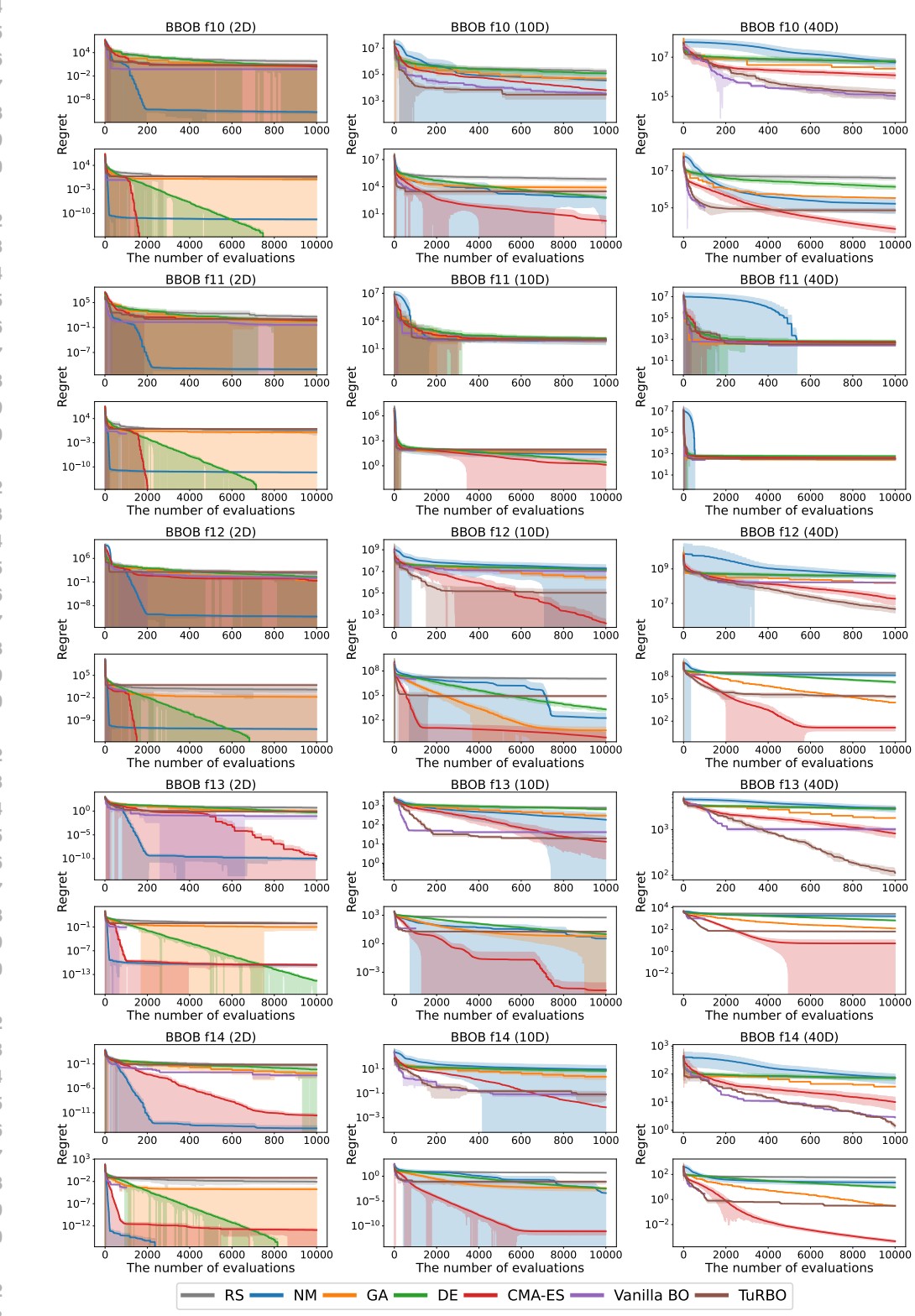

Figure 8: The convergence trajectories of the seven algorithms throughout the optimisation process on the BBOB f10-f14 with the low-dimension (left), medium-dimension (middle), and high-dimension (right). Each coloured line represents the mean regret (the difference between the true optimum and the best function value) obtained over 30 independent runs.

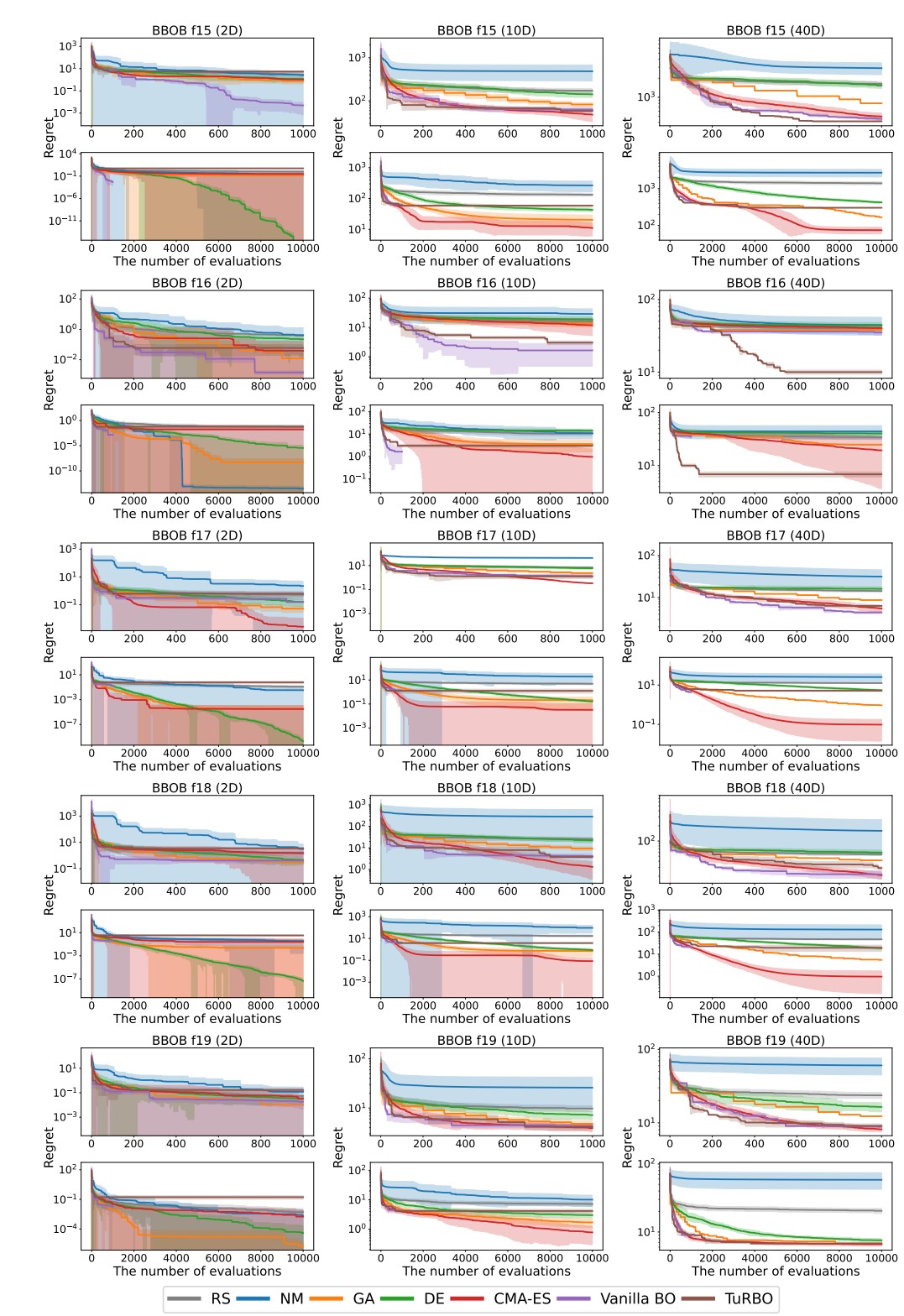

Figure 9: The convergence trajectories of the seven algorithms throughout the optimisation process on the BBOB f15-f19 with the low-dimension (left), medium-dimension (middle), and high-dimension (right). Each coloured line represents the mean regret (the difference between the true optimum and the best function value) obtained over 30 independent runs.

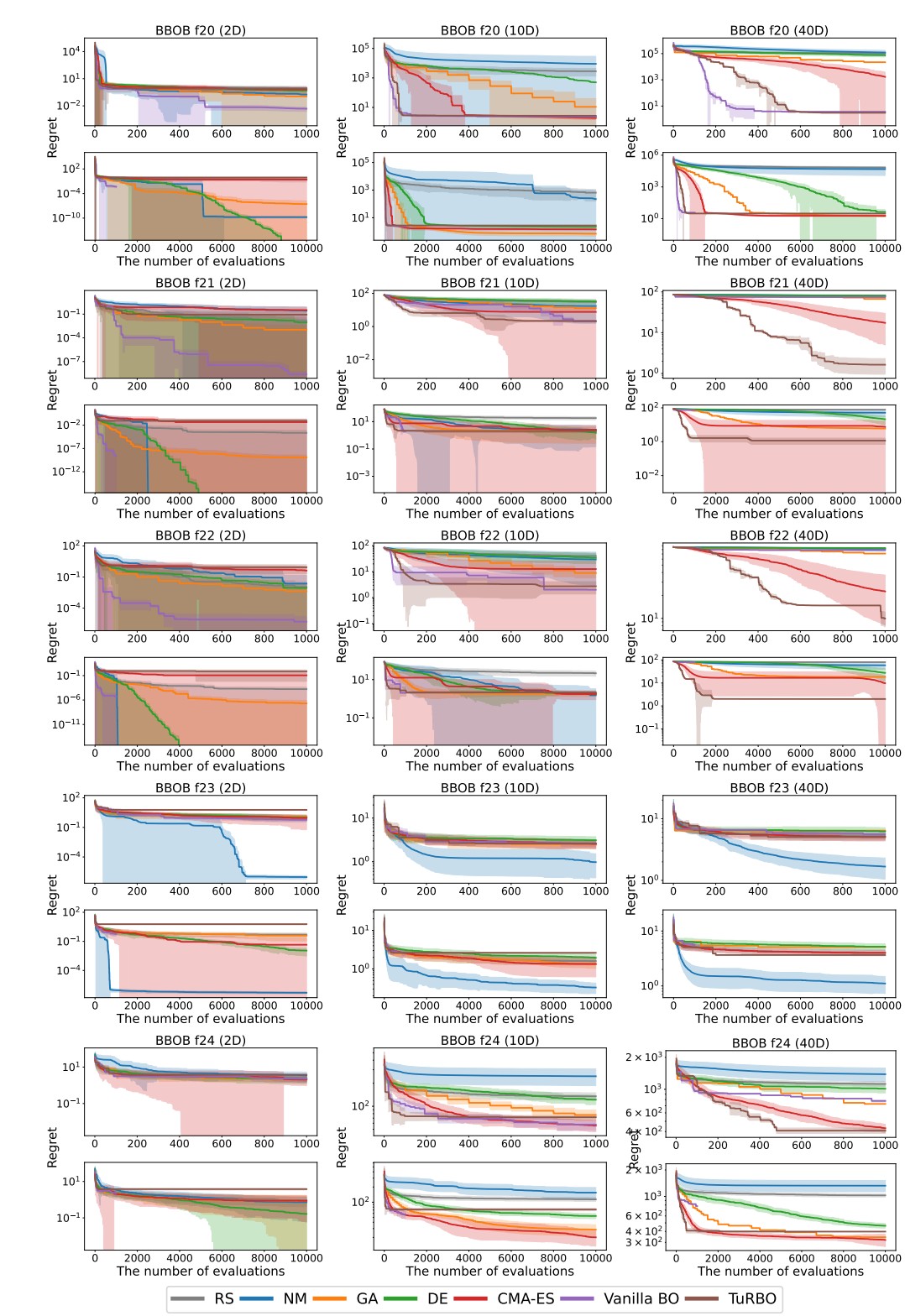

Figure 10: The convergence trajectories of the seven algorithms throughout the optimisation process on the BBOB f20-f24 with the low-dimension (left), medium-dimension (middle), and high-dimension (right). Each coloured line represents the mean regret (the difference between the true optimum and the best function value) obtained over 30 independent runs.

# C    LLM USAGE

We used large language models (LLMs) solely for language polishing.

