# OpenReview forum: "When to Use Which? An Investigation of Search Methods on Expensive Black-box Optimisation Problems"
_ICLR.cc/2026/Conference — Submitted to ICLR 2026_

### Official Review · Reviewer_rckW · 2025-10-15

**Soundness:** 1
**Presentation:** 3
**Contribution:** 1
**Rating:** 2
**Confidence:** 5

**Summary:**

This paper addresses the problem of choosing the right algorithm for black box optimisation problems. To do so, the authors consider a series of algorithms belonging to different paradigms (evolutionary algorithms, Bayesian optimisation, local search, etc.) and evaluate their performance on 24 BBOB functions.

**Strengths:**

S1. The paper addresses a legitimate but not particularly novel question in the world of optimisation: given a BBO, what is the most appropriate algorithm that a practitioner should use?
S2. The paper is very well written and easy to follow, I enjoyed reading it.

**Weaknesses:**

W1. The authors address the question through a fairly extensive experimental comparison and draw conclusions about the algorithm to be used based on the evaluation budget. However, this approach has a serious problem, in my view. How extrapolatable are the conclusions drawn about the 24 BBOB functions to any other real-world problem? Can the authors really guarantee that the recommendations they provide are accurate?

W2. (Continuing previous weakness) The authors are correct in the paper when they say, in line 137, ‘... the circumstances under which one algorithm surpasses another remain unclear.’ And this paper does not clarify anything in that regard. Although I am no expert on the subject, I understand that other approaches such as Runtime Analysis or Problem decomposition (Walsh, Fourier, Elementary landscapes) are usually much more effective (although perhaps not applicable here).

W3. I would suggest that the statistical analysis used to construct Table 1 is erroneous:
(1) Wilcoxon is for pairwise comparisons, whereas in this case there are multiple comparisons, which should be tested using Friedman tests.
(2) The assumptions for applying Wilcoxon are verified (the distributions of the algorithms should have the same shape).
(3) The way in which the test is applied is not clearly explained. Are fitness values from different instances mixed in the test? How is this implemented?
(4) We have no information on the magnitude of the difference in performance. This analysis is highly relevant and has been omitted.
(5) Did you run p-value corrections in the test when doing multiple comparisons?

**Questions:**

Please answer my questions in W1.

---

> ### Author Response · Authors · 2025-11-25
> **Response [1/2]**
>
> **W1: how extrapolatable are the conclusions drawn about the 24 BBOB functions to any other real-world problems?**
>
> Indeed, real-world problems exhibit a great deal of variety, and no benchmark suite can fully capture their complexity. However, they can in general be characterised based on a group of problem landscape features such as variable separability, conditioning, multi-modality, and global structure. The BBOB function benchmark is widely regarded as representatives of the typical difficulties that arise in real-world applications [Pál et al., 2012], and its 24 functions were explicitly designed to capture these features across diverse levels of separability, conditioning, multi-modality, and global structure [Hansen et al., 2009]. Indeed, BBOB functions have been demonstrated to exhibit diverse and heterogeneous landscape distributions that represent the variety in real-world problems [Long et al. 2022], and real-world optimisation tasks lie close to specific BBOB functions in feature space [Thomaser, et al., 2023], which leads to similar performances across different optimisation algorithms, as reported in [Schneider et al., 2022]. As such, we are confident that the recommendations found in this work are in general applicable to real-world optimisation problems.
>
>
> That said, in order to further address your concern (as well as those raised by Reviewers cVmS and Eke7), we have now included two additional real-world continuous black-box problems in hyperparameter optimisation, the SVM-HPO problem (2D) from HPOBench [Eggensperger et al., 2021] and the Lasso problem (10D) from LassoBench [Šehić et al., 2022]. The former one aims to minimise the validation loss and the latter is to minimise the mean squared error.
>
>
> Figure 5 in Section 4.4 in the revised manuscript (where we added a new section for real-world problems) presents the convergence trajectories of the seven algorithms throughout the search process on these two problems. As can be seen from the figure, the performance of the optimisation methods on these real-world problems is in general consistent with their performance on the BBOB benchmark (Sections 4.1-4.3). When the budget is very tight ($\le 200$), vanilla BO (purple line) performs best. As the budget increases, CMA-ES (red line) catches up, overtakes vanilla BO, and remain the top-performing method thereafter.
>
> It is worth noting that, interestingly, on the 10D Lasso problem, GA becomes slightly better than CMA-ES once the budget reaches around 2,000. A possible explanation is that, in the Lasso task, only 3 of the 10 variables substantially interact and contribute to the objective value, making the problem largely variable-separable [Šehić et al., 2022]. This aligns with the observation on the BBOB benchmark that GA works well on problems with separable variables (Section 4.3). This result indicates the benefit of considering problem characteristics in algorithm selection when such information is available.
>
>
>
> **W2: the paper does not clarify on “the circumstances under which one algorithm surpasses another remains unclear”. Other approaches such as runtime analysis or problem decomposition (Walsh, Fourier, Elementary landscapes) are usually much more effective (although perhaps not applicable here).**
>
>
> In black-box optimisation problems, the objective functions are not accessible. Consequently, runtime analysis or problem decomposition are typically not applicable, as they require explicit problem formulations to derive performance bounds or structural decompositions. As such, like other studies on black-box optimisation (e.g., [Muñoz and Smith-Miles, 2017; Ru et al., 2020]), we adopt an empirical approach in this work.

---

> > ### Author Response · Authors · 2025-11-25
> > **Response [2/2]**
> >
> > **W3: erroneous statistical analysis in Table 1: (1) Multiple comparisons should be tested using Friedman tests instead of Wilcoxon tests. (2) Assumptions of Wilcoxon. (3) The way in which the test is applied is not clearly explained. Are fitness values from different instances mixed? (4) No information on magnitude of the differences in performance. (5) Did you run p-value corrections in the test when doing multiple comparisons?**
> >
> > We address your comments regarding Table 1 one by one below.
> >
> > (1) Wilcoxon vs. Friedman:
> >
> >
> > While Friedman tests with post-hoc analysis are a valid approach for comparing multiple groups, our use of Wilcoxon rank-sum tests with Holm correction [Holm, 1979] follows established methodological guidelines for algorithm comparison [Derrac et al., 2011; Jia et al., 2025]. This approach controls family-wise error rate while enabling direct pairwise comparisons [Hollander et al., 2015], and it is widely adopted in many studies [Abualigah et al., 2022; Çimen et al., 2023; Li et al., 2024; Asklany et al., 2025].
> >
> >
> >
> > (2) Assumptions of the Wilcoxon test:
> >
> > In empirical comparisons of (optimisation) algorithms, although different methods may produce different distribution shapes [Jia et al., 2025], Wilcoxon remains widely used because it does not require identical shapes, but only that the performances of the algorithms are independent and ordinally comparable [Hollander et al., 2015]. It is statistically sound for assessing whether one algorithm outperforms another [Derrac et al., 2011; Jia et al., 2025].
> >
> >
> > (3) How the test is applied? Are fitness values from different instances mixed?
> >
> > The fitness values from different instances are not mixed. We apply the statistical tests separately for each of the 24 functions under a given evaluation budget (e.g., 200 evaluations), and then count the number of functions that each algorithm is statistically the best. More specifically, for a specific function (e.g., BBOB-f1), each algorithm yields a sample of 30 regret values. We then perform pairwise comparisons between algorithms using the Wilcoxon rank-sum test with Holm-Bonferroni correction to determine whether an algorithm is statistically the best for that specific function–budget setting (as we aim to find the best algorithm for recommendations). Finally, we count the number of functions on which each algorithm is statistically the best under that budge, and summarise these results in Table 1.
> >
> >
> >
> > (4) Magnitude of differences:
> >
> > We agree that reporting the magnitude of performance differences is useful. In the revised paper, on top of the mean and standard deviation of the regret values for all algorithms (Tables 3–6 in Appendix B.1), we will add the Vargha-Delaney's $A_{12}$ between algorithm pairs (a common non-parametric effect size measure for algorithm performances [Derrac et al., 2011]) to give readers a clearer sense of the practical magnitude of differences.
> >
> > (5) Did you run p-value corrections?
> >
> > Yes, we apply the Holm–Bonferroni correction [Holm, 1979] to adjust p-values across pairwise comparisons within each function–budget pair (Section 3.2, line 192).
> >
> > To sum up, in the revised paper we will incorporate the reviewer's suggestions and/or clarify possible misunderstandings regarding the statistical testing. Thank you for pointing out these issues.

---

> > > ### Author Response · Authors · 2025-11-25
> > > **References**
> > >
> > > **References**
> > >
> > >
> > > [Pál et al., 2012] Pál, L., Csendes, T., Markót, M.C., Neumaier, A., 2012. Black box optimization benchmarking of the global method. Evolutionary Computation, 2012.
> > >
> > > [Hansen et al., 2009] Hansen, N., Finck, S., Ros, R. and Auger, A., 2009. Real-parameter black-box optimization benchmarking 2009: Noiseless functions definitions. INRIA Report, 2009.
> > >
> > > [Long et al. 2022] Long, F.X., Vermetten, D., Stein, B.B., Kononova, A.V., 2022. BBOB instance analysis: Landscape properties and algorithm performance across problem instances. EvoStar, 2022.
> > >
> > > [Thomaser, et al., 2023] Thomaser, A., Vogt, ME., Kononova, A.V., Bäck, T., 2023. Transfer of multi-objectively tuned CMA-ES parameters to a vehicle dynamics problem. EMO, 2023.
> > >
> > > [Schneider et al., 2022] Schneider, L., Schäpermeier, L., Prager, R.P., Bischl, B., Trautmann, H., Kerschke, P., 2022. HPO $\times$ ELA: Investigating hyperparameter optimization landscapes by means of exploratory landscape analysis. PPSN, 2022.
> > >
> > > [Eggensperger et al., 2021] Eggensperger, K., Müller, P., Mallik, N., Feurer, M., Sass, R., Klein, A., Awad, N., Lindauer, M. and Hutter, F., 2021. HPOBench: A collection of reproducible multi-fidelity benchmark problems for HPO. NeurIPS, 2021.
> > >
> > > [Šehić et al., 2022] Šehić, K., Gramfort, A., Salmon, J. and Nardi, L., 2022. Lassobench: A high-dimensional hyperparameter optimization benchmark suite for lasso. AutoML, 2022.
> > >
> > > [Muñoz and Smith-Miles, 2017] Muñoz, M. A. and Smith-Miles, K.A., 2017. Performance analysis of continuous black-box optimization algorithms via footprints in instance space. Evolutionary Computation, 2017.
> > >
> > > [Ru et al., 2020] Ru, B., Alvi, A.S., Nguyen, V., Osborne, M.A. and Roberts, S.J., 2020. Bayesian optimisation over multiple continuous and categorical inputs. ICML, 2020.
> > >
> > > [Holm, 1979] Holm, S., 1979. A simple sequentially rejective multiple test procedure. Scandinavian Journal of Statistics, 1979.
> > >
> > > [Derrac et al., 2011] Derrac, J., García, S., Molina, D., and Herrera, F., 2011. A practical tutorial on the use of nonparametric statistical tests as a methodology for comparing evolutionary and swarm intelligence algorithms. Swarm and Evolutionary Computation, 2011.
> > >
> > > [Jia et al., 2025] Jia, B., Liu, J. and Zhang, M., 2025. Pairwise statistical comparisons of multiple algorithms. Frontiers of Computer Science, 2025.
> > >
> > > [Hollander et al., 2015] Hollander, M., Wolfe, D. A., and Chicken, E., 2015. Nonparametric Statistical Methods. Wiley, 2015.
> > >
> > > [Abualigah et al., 2022] Abualigah, L., Elaziz, M.A., Khasawneh, A.M. et al., 2022. Meta-heuristic optimization algorithms for solving real-world mechanical engineering design problems: a comprehensive survey, applications, comparative analysis, and results. Neural Comp. and Applic., 2022.
> > >
> > > [Çimen et al., 2023] Çimen, M.E., Garip, Z., Boz, A.F., 2023. Comparison of metaheuristic optimization algorithms for numerical solutions of optimal control problems. Concurrency Computat. Pract. Exper., 2023.
> > >
> > > [Li et al., 2024] Li, M., Han, X., Chu, X. and Liang, Z., 2024. Empirical comparison between MOEAs and local search on multi-objective combinatorial optimisation problems. GECCO, 2024.
> > >
> > > [Asklany et al., 2025] Asklany, A.M., Tawhid, M.A., El-Hawary, H.M. and Allam, A.A., 2025. Comparative analysis of meta-heuristic algorithms for unconstrained optimization problems. Journal of Multidisciplinary Scientific Research, 2025.

---

> > > > ### Comment · Reviewer_rckW · 2025-11-26
> > > >
> > > > Thank you for your response. The authors have made a significant effort to address the issues raised and improve the paper in that regard.
> > > >
> > > > I acknowledge the effort made to include two “real” problems in order to better support the conclusions. However, I remain unconvinced with respect to W1. The fact that the paper is purely experimental, probably due to the lack of access to the objective function of the problem, means that the conclusions drawn are superficial.
> > > >
> > > > The authors say that the 24 BBOB functions are diverse and somehow cover the different possibilities, being sufficiently representative for the analyses carried out on them to be relevant in real-world contexts. I cannot say that they are not diverse, but it is very difficult for the authors to claim that the recommendations observed in this work are “generally applicable to real-world problems.” And I think this aspect is critical in determining the relevance of the paper in general.
> > > >
> > > > Following this line of thought, it seems to me that the recommendations put forward lack depth in their reasoning. I am sorry to say that I do not see the contribution of the work.

---

> ### Author Response · Authors · 2025-11-30
>
> Thank you for your prompt reply. However, we respectfully do not agree with the reviewer's opinion that findings obtained from empirical studies on benchmark problems lack depth in reasoning and cannot, in general, be applied to real-world problems. If this were the case, it would call into question the value of testing, evaluating and comparing algorithms on benchmark problems, a practice that is widely accepted and fundamental in the optimisation field.
>
> Despite the vast variety of real-world optimisation problems, they can generally be characterised based on a set of landscape features such as variable separability, conditioning, multi-modality, and global structure. The BBOB benchmark was explicitly designed based on this principle, with its 24 functions systematically capturing these features across varying degrees of separability, conditioning, multi-modality, and global structure. Numerous studies (e.g., Pál et al., 2012; Long et al., 2023; Thomaser et al., 2023) have shown that BBOB functions effectively represent the key landscape features that govern algorithmic performance on real-world optimisation problems, thereby providing a robust approximation of the diversity encountered in practical applications. As such, we are confident that the findings obtained in this work are in general applicable to practical real-world optimisation settings.
>
>
> **References**
>
> [Pál et al., 2012] Pál, L., Csendes, T., Markót, M.C., Neumaier, A., 2012. Black box optimization benchmarking of the global method. Evolutionary Computation, 2012.
>
> [Long et al. 2023] Long, F.X., Vermetten, D., Stein, B.B., Kononova, A.V., 2023. BBOB instance analysis: Landscape properties and algorithm performance across problem instances. EvoStar, 2023.
>
> [Thomaser et al., 2023] Thomaser, A., Vogt, ME., Kononova, A.V., Bäck, T., 2023. Transfer of multi-objectively tuned CMA-ES parameters to a vehicle dynamics problem. EMO, 2023.

---

### Official Review · Reviewer_vV8X · 2025-10-31

**Soundness:** 3
**Presentation:** 3
**Contribution:** 2
**Rating:** 6
**Confidence:** 4

**Summary:**

The paper compares a range of standard black-box optimization problems on a group of standard test problems to investigate which algorithm is best applied under what circumstances, including the evaluation budget, problem dimensionality and objective characteristics.

**Strengths:**

The comparisons appear fair and objective. While there is nothing revolutionary here, I feel that this paper is helpful in providing a high-level overview of the field for users. Results are fairly thorough, and more-or-less reflect what I would expect (caveat: this may be confirmation bias as I work in BO research).

**Weaknesses:**

One fairly important weakness of this paper is a lack of detail on the BO algorithm settings (and, presumably, the other algorithms, though I am less familiar with these). I assume, based on the sparsity of information, that the authors have simply used whatever settings (acquisition function, GP kernel etc) are provided out of the box. While this is not necessarily unreasonable (many - perhaps even most - users will apply BO precisely like this, for better or worse) I feel it is important to at least describe these settings. For example is "vanilla-BO" using EI (expected improvement)?  GP-UCB (GP upper conficence bound in one of its many variants)? Thompson sampling? I'm also guessing the kernel (covariance) is an SE kernel with lengthscale tuned for log-likelihood, based both on experience and comments on line 362, but this really needs to be defined explicitly.

One other (related) point of weakness is section 4.3. In BO, knowledge of problem characteristics is often used to achieve a substantial speedup (or just avoid a slowdown). In particular kernel selection should reflect knowledge of the problem domain - eg smooth functions are well suited to an SE kernel, whereas if you have "sharp edges" then a (lower-order) Matern may be preferred; separable variables may lead you to prefer a Laplacian kernel or a product of kernels on different axis; and awareness of local structure may lead you to use some variation of kernel localization. Some discussion of these issues may be helpful.

**Questions:**

See weaknesses.

---

> ### Author Response · Authors · 2025-11-25
> **Response**
>
> **W1: detail of settings of BO and other optimisation algorithms**
>
> We should have been clearer about this. Yes, exactly, we use the default settings out of the box because our goal is to provide guidance for practitioners who are not specialist in optimisation, helping them select off-the-shelf black-box optimisers according to their budgets.
>
> Specifically, in vanilla BO, we use the commonly used expected improvement acquisition function together with the RBF kernel from BoTorch's default implementation. For TuRBO, we follow the authors' recommendations and employ Thompson Sampling and a Matérn 5/2 kernel, and we adopt a batch size of 5 (following the practice in [Santoni et al. 2024]). In both BO methods, kernel lengthscales are learned by maximising the marginal log-likelihood. The settings for BO and the other optimisation algorithms are provided in Appendix A.1 (Method Details). We will explicitly include all relevant parameter details in the revised paper.
>
>
> **W2: in BO, knowledge of problem characteristics, particularly kernel selection, is often used to achieve a substantial speedup**
>
>
> Very good point. Indeed, under different properties of the BBOB benchmark (e.g., smoothness, separability and multi-modality), using different kernel families can substantilly improve the performance of BO algorithms. For example, SE kernels are well suited to very smooth functions; lower-order Matérn kernels can better capture sharp or rough (non-smooth) behaviour; a product of kernels or additive kernels can exploit separability; and kernel localisation can better handle multi-modal functions with high-quality local optima. Although the aim of this work is to investigate off-the-shelf optimisers using their default settings, it is important to point out potential improvement when some problem characteristics are accessible. We will add a discussion of this point in the revised paper, as suggested.
>
> **References**
>
> [Santoni et al. 2024] Santoni, M.L., Raponi, E., Leone, R.D. and Doerr, C., 2024. Comparison of high-dimensional Bayesian optimization algorithms on BBOB. ACM Transactions on Evolutionary Learning and Optimization, 2024.

---

### Official Review · Reviewer_Eke7 · 2025-10-31

**Soundness:** 3
**Presentation:** 4
**Contribution:** 3
**Rating:** 8
**Confidence:** 5

**Summary:**

This paper investigates which optimization algorithm is most appropriate for expensive black-box optimization (BBO) problems under varying evaluation budget constraints. The authors consider common families of search methods – including Bayesian optimization (BO) with Gaussian processes, an advanced BO variant (TuRBO), evolutionary algorithms (CMA-ES, Genetic Algorithm, Differential Evolution), a local search method (Nelder–Mead), and random search – and compare their performance across different budget regimes and problem characteristics. The main contribution is an extensive empirical study that provides practical guidelines on “when to use which” optimizer, depending on how tight the evaluation budget is and what is known about the problem’s landscape.

**Strengths:**

- The study is exceptionally thorough in its empirical scope. It tests seven distinct algorithms (random search, Nelder–Mead, GA, DE, CMA-ES, vanilla BO, and TuRBO) across a wide range of conditions. Experiments cover 24 benchmark functions with diverse properties, multiple budget levels (200, 800, 2,000, and 8,000 evaluations, plus continuous trajectories up to 10,000), and different dimensionalities (2D, 10D, 40D).
- The paper employs appropriate statistical tests to support its claims. For each benchmark function and budget, it identifies the best algorithm using Wilcoxon rank-sum tests with Holm–Bonferroni correction.
- Practical Guidance and Clear Takeaways: for example, they conclude vanilla BO is the default choice under very tight evaluation budgets, CMA-ES is best when the budget is sufficiently large, and methods like NM or TuRBO become preferable in certain niche scenarios

**Weaknesses:**

- All experiments are conducted on the BBOB artificial test functions, which, while standard in optimization research, are still simplified representations of real-world tasks. IMHO, the more interesting question is the actual computation time of each optimization method, e.g., wall-clock time, when considering the *actual* computation cost/time of each function evaluation. For instance, if the time complexity of each function evaluation is less than O(n^3) - the time complexity of BO, then its advantage in the tight budget setting is questionable.
- Overall, the paper is well-written, but a few minor points could be improved. For example, some of the terminology like “weak global structure” or “adequate global structure” (from the BBOB categorization) might be unclear to readers unfamiliar with those benchmarks.
-

**Questions:**

- Could you clarify why certain popular black-box optimizers were not included in the study? For instance, Particle Swarm Optimization (PSO) is a well-known heuristic for continuous optimization – was it omitted for any specific reason?
- How confident are the authors that the findings will generalize to real-world expensive optimization tasks? The benchmarks used are synthetic and noise-free. For a practical problem,  say, optimizing hyperparameters of a machine learning model, which might have noise and unknown constraints, would the recommendation still be “use BO for ≤200 evaluations, CMA-ES for ≥1000,” etc.?
- Section 4.3 groups problems by characteristics (separability, conditioning, multi-modality) and gives algorithm recommendations per category. In a real scenario, a user may not know these properties of their objective function upfront. What do the authors suggest for a practitioner to identify, for example, that their problem has a “weak global structure” or is “highly conditioned”? This is important if the user wants to apply the paper's advice. Any guidance on how to assess problem features in practice would enhance the utility of those recommendations.

---

> ### Author Response · Authors · 2025-11-25
> **Response [1/2]**
>
> **Q1: why certain popular black-box optimisers (e.g. PSO) were not included?**
>
> In this study, we consider seven representative optimisers (Random Search, Nelder–Mead, BO, TuRBO, GA, DE and CMA-ES) as they cover the canonical and widely studied families of optimisers commonly used in expensive black-box optimisation. These include one baseline (Random Search), one local search optimiser (Nelder–Mead), two BO-based optimisers, and three evolutionary computation (EC) optimisers. Among the EC optimisers, CMA-ES is an explicitly model-based optimiser, whereas GA and DE are implicitly model-based. Between GA and DE, GA generally favours broader global exploration, whereas DE tends to converge faster. PSO, similar to DE, is an implicitly model-based optimiser, which often achieves faster convergence than GA but with weaker global exploration ability. Existing work has shown that PSO can exhibit search behaviour highly similar to that of DE [Chen et al., 2019; Kachitvichyanukul, 2012]. As such, we did not include PSO in our study. That said, we fully acknowledge the distinctions among different black-box optimisers and will clarify this point in the revised paper.
>
>
> **Q2\&W1, part1: while standard in optimisation research, BBOB functions are still simplified representations of real-world tasks. How well the findings generalise to real-world expensive optimisation tasks?**
>
>
> Despite the vast variety of real-world problems, they can in general be characterised based on a group of problem landscape features such as variable separability, conditioning, multi-modality, and global structure.
> The BBOB function benchmark is widely regarded as representatives of the typical difficulties that arise in real-world applications [Pál et al., 2012], and its 24 functions were explicitly designed to capture these features across diverse levels of separability, conditioning, multi-modality, and global structure) [Hansen et al., 2009].
> Indeed, BBOB functions have been demonstrated to exhibit diverse and heterogeneous landscape distributions that represent the variety in real-world problems [Long et al. 2022], and real-world optimisation tasks lie close to specific BBOB functions in feature space [Thomaser et al., 2023], which leads to similar performances across different optimisation algorithms, as reported in [Schneider et al., 2022].
>
> That said, in order to further address your concern (as well as those raised by Reviewers cVmS and rckW), we have now included two additional real-world continuous black-box problems in hyperparameter optimisation, the SVM-HPO problem (2D) from HPOBench [Eggensperger et al., 2021] and the Lasso problem (10D) from LassoBench [Šehić et al., 2022]. The former one aims to minimise the validation loss and the latter is to minimise the mean squared error.
>
>
> Figure 5 in Section 4.4 in the revised manuscript (where we added a new section for real-world problems) presents the convergence trajectories of the seven algorithms throughout the search process on these two problems. As can be seen from the figure, the performance of the optimisation methods on these real-world problems is in general consistent with their performance on the BBOB benchmark (Sections 4.1-4.3). When the budget is very tight ($\le 200$), vanilla BO (purple line) performs best. As the budget increases, CMA-ES (red line) catches up, overtakes vanilla BO, and remain the top-performing method thereafter.
>
> It is worth noting that, interestingly, on the 10D Lasso problem, GA becomes slightly better than CMA-ES once the budget reaches around 2,000. A possible explanation is that, in the Lasso task, only 3 of the 10 variables substantially interact and contribute to the objective value, making the problem largely variable-separable [Šehić et al., 2022]. This aligns with the observation on the BBOB benchmark that GA works well on problems with separable variables (Section 4.3). This result indicates the benefit of considering problem characteristics in algorithm selection when such information is available.

---

> > ### Author Response · Authors · 2025-11-25
> > **Response [2/2]**
> >
> > **Q2, part2: how can the recommendations be applied under noise and unknown constraints (e.g., BO for $\leq 200$ evaluations and CMA-ES for $\geq 1000$)?**
> >
> >
> > The applicability of the recommendations is closely related to how well optimisation algorithms can accommodate specific problem features.
> > For noisy problems, BO is naturally suitable because Gaussian process models explicitly account for observational noise, and the acquisition function works with the resulting posterior distribution. We hence firmly believe the recommendation that BO is best for $\le 200$ evaluations holds under noise environments.
> > CMA-ES is also generally robust to noise, as its distribution-based updates inherently smooth noise effects, though strong noise may require dedicated noise-handling variants [Nomura et al., 2023]. Similarly, we believe the recommendation that CMA-ES works well under generous budgets ($\ge 800$) still holds for noise problems.
> >
> > As for problems with unknown constraints, optimisation algorithms may spend part of the budget evaluating infeasible solutions, resulting in fewer valid data to build the model of finding high-quality solutions. As such, the recommendations may shift slightly to the ``right''; for example, the budget range in which BO works generally best may extend beyond 200 evaluations.
> >
> >
> >
> > **Q3: Section 4.3, guidance on how to identify problem characteristics (separability, conditioning, multi-modality, global structure) in practice?**
> >
> > Indeed, in many BBO practical problems, the user may have little idea about the problem's characteristics. However, in some situations the user may have some sense of them based on domain knowledge. For example, in software configuration tuning, the systems generally have many interacting toggles and conditional dependencies, making the search space discretised and hard to predict the search direction [Huang et al., 2022]. In such scenarios, there typically exist many similar local optima and a weak global structure [Chen and Li, 2021]. A counter-example is the routing problem within a map where cities are regularly distributed (resembling a Euclidean TSP), where high-quality tours typically share many edges, which creates a predictable ``big valley’’ trend [Hains et al., 2011], hence an adequate global structure.
> >
> > Regarding variable separability, in rocket engine mass-flow modelling, variables (e.g., stagnation pressure, temperature and heat-capacity ratio) influence mass flow through distinct mechanisms. Engineers therefore expect them to be separable [Chen et al., 2017]. Another example is the portfolio optimisation of a large hedge fund, where individual assets contribute additively to total return [Moehle et al., 2023].
> >
> > As for conditioning, in structural finite element models (e.g., minimising compliance or weight of a bridge, subject to stiffness requirements), the stiffness matrix often has eigenvalues spanning many orders of magnitude. This leads to large disparity in the optimisation landscape and thus a high condition number (e.g., over $10^8$ [Kannan, 2014]). In contrast, when optimising movements control in animation production subject to limited action sequences, such short planning enforces roughly uniform sensitivity across movement directions, which leads to a low condition [Hämäläinen et al., 2022].
> >
> > We will add a discussion about this issue - thank you for this helpful comment.
> >
> >
> >
> >
> > **W1, part2: actual computation time of each optimisation method against actual computation cost of each function evaluation**
> >
> > Very good point. Indeed, when taking into account the actual computation cost of an optimisation method, i.e., the time required to evaluate a solution plus the algorithm's own overhead to identify that solution, the recommendations may change. For example, when the available time is very limited, such as a few seconds (e.g., optimisation in online setting), the advantage of BO becomes less pronounced. In contrast, methods whose per-iteration computational cost is independent of the number of solutions evaluated become more suitable, such as EAs. We will include a discussion of this point in the revised paper.
> >
> >
> >
> > **W2: some terminologies like weak/adequate global structure might be unclear to readers**
> >
> > We will provide explanations of all relevant terminology in the revised paper.

---

> > > ### Author Response · Authors · 2025-11-25
> > > **References**
> > >
> > > **References**
> > >
> > > [Chen et al., 2019] Chen, S., Bolufé-Röhler, A., Montgomery, J. and Hendtlass, T., 2019, June. An analysis on the effect of selection on exploration in particle swarm optimization and differential evolution. CEC, 2019.
> > >
> > > [Kachitvichyanukul, 2012] Kachitvichyanukul, V., 2012. Comparison of three evolutionary algorithms: GA, PSO, and DE. Industrial Engineering and Management Systems, 2012.
> > >
> > > [Pál et al., 2012] Pál, L., Csendes, T., Markót, M.C., Neumaier, A., 2012. Black Box Optimization Benchmarking of the GLOBAL Method. Evolutionary Computation, 2012.
> > >
> > > [Hansen et al., 2009] Hansen, N., Finck, S., Ros, R. and Auger, A., 2009. Real-parameter black-box optimization benchmarking 2009: Noiseless functions definitions. INRIA Report, 2009.
> > >
> > > [Long et al. 2022] Long, F.X., Vermetten, D., Stein, B.B., Kononova, A.V., 2022. BBOB instance analysis: Landscape properties and algorithm performance across problem instances. EvoStar, 2022.
> > >
> > > [Thomaser et al., 2023] Thomaser, A., Vogt, ME., Kononova, A.V., Bäck, T., 2023. Transfer of multi-objectively tuned CMA-ES parameters to a vehicle dynamics problem. EMO, 2023.
> > >
> > > [Schneider et al., 2022] Schneider, L., Schäpermeier, L., Prager, R.P., Bischl, B., Trautmann, H., Kerschke, P., 2022. HPO $\times$ ELA: Investigating hyperparameter optimization landscapes by means of exploratory landscape analysis. PPSN, 2022.
> > >
> > > [Eggensperger et al., 2021] Eggensperger, K., Müller, P., Mallik, N., Feurer, M., Sass, R., Klein, A., Awad, N., Lindauer, M. and Hutter, F., 2021. HPOBench: A collection of reproducible multi-fidelity benchmark problems for HPO. NeurIPS, 2021.
> > >
> > > [Šehić et al., 2022] Šehić, K., Gramfort, A., Salmon, J. and Nardi, L., 2022. Lassobench: A high-dimensional hyperparameter optimization benchmark suite for lasso. AutoML, 2022.
> > >
> > > [Nomura et al., 2023] Nomura, M., Akimoto, Y. and Ono, I., 2023, July. CMA-ES with learning rate adaptation: Can CMA-ES with default population size solve multimodal and noisy problems?. GECCO, 2023.
> > >
> > > [Huang et al., 2022] Huang, S., Wu, X., Finkel, H., et al., 2022. Unlocking the secrets of software configuration: A fitness-landscape study of real-world systems. arXiv, 2022.
> > >
> > > [Chen and Li, 2021] Chen T., Li M., 2021. Multi-objectivizing software configuration tuning. In FSE, 2021.
> > >
> > > [Hains et al., 2011] Hains, D., Whitley, L. and Howe, A., 2011. Revisiting the big valley search space structure in the TSP. J Oper Res Soc, 2011.
> > >
> > > [Chen et al., 2017] Chen, X., Zhang, H., Lin, S., and Wong, K, 2017. Fast modeling methods for complex system with separable features. arXiv, 2017.
> > >
> > > [Moehle et al., 2023] Moehle, N., Gindi, J., Boyd, S. et al., 2023. Portfolio construction as linearly constrained separable optimization. Optimisation and Engineering, 2023.
> > >
> > > [Kannan, 2014] Kannan, R., Hendry, S., Higham, N.J. and Tisseur, F., 2014. Detecting the causes of ill-conditioning in structural finite element models. Computers and Structures, 2014.
> > >
> > > [Hämäläinen et al., 2022] Hämäläinen, P., Toikka, J., Babadi, A. and Liu, C.K., 2022. Visualizing movement control optimization landscapes. IEEE Trans Vis Comput Graph, 2022.

---

### Official Review · Reviewer_cVmS · 2025-11-04

**Soundness:** 4
**Presentation:** 4
**Contribution:** 2
**Rating:** 4
**Confidence:** 4

**Summary:**

This paper studies the problem of determining which blackbox optimization method to use given a blackbox optimization problem. There are a wide variety of blackbox optimization methods developed under different sets of assumptions and for different evaluation budgets. For instance evolutionary algorithm based approaches typically work well for settings with a large budget while bayesian optimization methods work well for more budget constrained settings.

This paper studies this problem and provides guidance on which methods work best for which settings. Overall BO based methods work best for lower budgets while CMA-ES works best for higher budgets.

**Strengths:**

- This paper presents an extensive study on the effectiveness of BO and Evolutionary algorithms for black box optimization for various budget ranges.
- Various aspects of the black box optimization problem are considered such as dimensionality, multi-modality and other problem characteristics that can help make a better judgement of the best method.
- Through experiments on benchmark datasets, the best settings are determined in a data-driven manner.

**Weaknesses:**

- The main weakness of this paper is that all the runs are on BBOB functions, and no real world functions were considered. The findings from the study have not been applied to any real world blackbox optimization functions.
- From a practical viewpoint, it has been known for a long time that evolutionary algorithms perform well for higher budgets and bayesian optimization based approaches for lower budgets. The novelty of this study is low.
- While this paper empirically quantifies that BO works better at lower budgets, and evolutionary methods at higher budgets, it is unclear what practical benefits can this bring. Can this approach be used to build a meta algorithm which selects the optimization method given a problem? How much better would that algorithm be compared to any individual method. This has not been evaluated on any practical real world problems.

**Questions:**

(see above comments for details) Can this approach be used to build a meta algorithm which selects the optimization method given a problem? How much better would that algorithm be compared to any individual method.
How can one use the finding from this study to make BBO better?

---

> ### Author Response · Authors · 2025-11-25
> **Response [1/2]**
>
> **W1: no real-world problems are considered; all the runs are on BBOB functions**
>
> Despite the vast variety of real-world problems, they can in general be characterised based on a group of problem landscape features such as variable separability, conditioning, multi-modality, and global structure. The BBOB function benchmark is widely regarded as representatives of the typical difficulties that arise in real-world applications [Pál et al., 2012], and its 24 functions were explicitly designed to capture these features across diverse levels of separability, conditioning, multi-modality, and global structure [Hansen et al., 2009]. Indeed, BBOB functions have been shown to exhibit diverse and heterogeneous landscape distributions that represents the variety in real-world problems [Long et al. 2022], and real-world optimisation tasks lie close to specific BBOB functions in feature space [Thomaser et al., 2023], which leads to similar performances across different optimisation algorithms, as reported in [Schneider et al., 2022].
>
>
> That said, in order to further address your concern (as well as Reviewers Eke7's and rckW's), we have now included two additional real-world continuous black-box problems in hyperparameter optimisation, the SVM-HPO problem (2D) from HPOBench [Eggensperger et al., 2021] and the Lasso problem (10D) from LassoBench [Šehić et al., 2022]. The former one aims to minimise the validation loss, and the latter is to minimise the mean squared error.
>
> Figure 5 in Section 4.4 in the revised manuscript (where we added a new section for real-world problems) presents the convergence trajectories of the seven algorithms throughout the search process on these two problems. As can be seen from the figure, the performance of the optimisation methods on these real-world problems is in general consistent with their performance on the BBOB benchmark (Sections 4.1--4.3). When the budget is very tight ($\le 200$), vanilla BO (purple line) performs best. As the budget increases, CMA-ES (red line) catches up, overtakes vanilla BO, and remain the top-performing method thereafter.
>
> It is worth noting that, interestingly, on the 10D Lasso problem, GA becomes slightly better than CMA-ES once the budget reaches around 2,000. A possible explanation is that, in the Lasso task, only 3 of the 10 variables substantially interact and contribute to the objective value, making the problem largely variable-separable [Šehić et al., 2022]. This aligns with the observation on the BBOB benchmark that GA works well on problems with separable variables (Section 4.3). This result indicates the benefit of considering problem characteristics in algorithm selection when such information is available.
>
>
> **W2: it is long known that evolutionary algorithms perform well for higher budgets and Bayesian optimisation for lower budgets**
>
> Indeed, it has long been believed that BO performs better than EAs in tight budgets, and vice versa. However, interestingly, recent studies, e.g., [Ozaki et al., 2022; Daulton et al., 2022; Papenmeier et al., 2022], paint a more nuanced picture, showing EAs can do well in low budgets, while BO has potential under high budgets as well. For example, CMA-ES has been reported to outperform BO in hyperparameter optimisation of convolutional neural networks for image classification when the evaluation budget is very low (around 100 evaluations) [Ozaki et al., 2022]. Conversely, BO has been reported to achieve better performance than CMA-ES for a trajectory planning problem with a budget of 2,000 evaluations [Daulton et al., 2022].
>
> More importantly, it remains unclear that under what specific small-budget conditions BO is better than EAs, and what the ``comfort zones'' of different algorithms (BO, EAs, NM) are. Therefore, it is useful to provide clear, general guidance to practitioners on selecting among these popular off-the-shelf algorithms based on their available budgets and problem attributes (e.g., dimensionality).

---

> > ### Author Response · Authors · 2025-11-25
> > **Response [2/2]**
> >
> > **W3\&Q1: can this approach be used to build a meta-algorithm which selects the optimisation method given a problem? How much better would that algorithm be compared to any individual method?**
> >
> > Insightful comment. The aim of this work is to identify, among classical black-box optimisation methods, which one works best under different levels of budget tightness. In our view, it is not straightforward to directly construct a meta-algorithm that consistently outperforms the best individual method at every budget level. Having that said, for an individual method, making use of budget information and problem attributes could potentially improve its search. For example, in our results, vanilla BO has been shown to perform best when the budget is below 200 evaluations. Within this range, depending on the user's exact budget, one could design a BO variant that adjusts the exploration-exploitation balance according to the budget and problem dimensionality. That is, when the budget is extremely tight (e.g., around 50 evaluations) or when dimensionality is high, placing more emphasis on exploitation, and vice versa. We will add these discussions in the revised paper. Thank you for this thoughtful comment.
> >
> >
> > **References**
> >
> > [Pál et al., 2012] Pál, L., Csendes, T., Markót, M.C., Neumaier, A., 2012. Black box optimization benchmarking of the global method. Evolutionary Computation, 2012.
> >
> > [Hansen et al., 2009] Hansen, N., Finck, S., Ros, R. and Auger, A., 2009. Real-parameter black-box optimization benchmarking 2009: Noiseless functions definitions. INRIA Report, 2009.
> >
> > [Long et al. 2022] Long, F.X., Vermetten, D., Stein, B.B., Kononova, A.V., 2022. BBOB instance analysis: Landscape properties and algorithm performance across problem instances. EvoStar, 2002.
> >
> > [Thomaser et al., 2023] Thomaser, A., Vogt, ME., Kononova, A.V., Bäck, T., 2023. Transfer of multi-objectively tuned CMA-ES parameters to a vehicle dynamics problem. EMO, 2023.
> >
> > [Schneider et al., 2022] Schneider, L., Schäpermeier, L., Prager, R.P., Bischl, B., Trautmann, H., Kerschke, P., 2022. HPO $\times$ ELA: Investigating hyperparameter optimization landscapes by means of exploratory landscape analysis. PPSN, 2022.
> >
> > [Eggensperger et al., 2021] Eggensperger, K., Müller, P., Mallik, N., Feurer, M., Sass, R., Klein, A., Awad, N., Lindauer, M. and Hutter, F., 2021. HPOBench: A collection of reproducible multi-fidelity benchmark problems for HPO. NeurIPS, 2021.
> >
> > [Šehić et al., 2022] Šehić, K., Gramfort, A., Salmon, J. and Nardi, L., 2022. Lassobench: A high-dimensional hyperparameter optimization benchmark suite for lasso. AutoML, 2022.
> >
> > [Ozaki et al., 2022] Ozaki, Y., Takenaga, S. and Onishi, M., 2022. Global search versus local search in hyperparameter optimization. CEC, 2022.
> >
> > [Daulton et al., 2022] Daulton, S., Eriksson, D., Balandat, M. and Bakshy, E., 2022. Multi-objective Bayesian optimization over high-dimensional search spaces. UAI, 2022.
> >
> > [Papenmeier et al., 2022] Papenmeier, L., Nardi, L. and Poloczek, M., 2022. Increasing the scope as you learn: Adaptive Bayesian optimization in nested subspaces. NeurIPS, 2022.

---

### Meta-Review · Area_Chair_sUmE · 2026-01-10

**Summary:**

This paper tries to empirically address the following question: given a black-box optimization problem and associated specification (e.g., evaluation budget), which algorithm among a given library should a practitioner use. Through empirical evaluation on 24 benchmarks, the paper provides guidance on which methods work best for which settings: typically, BO based methods work best for lower budgets while CMA-ES works best for higher budgets.

All the reviewers' appreciated the studied question, writing of the paper, and general execution. However, they raised some important concerns:
1. Lack of experiments on real-world domains
2. Applicability of the recommendations when some of the problem characteristics are not known
3. Statistical tests performed for comparing algorithms

The authors have responded by acknowledging and addressing #3. To address #1, the authors have included two simple hyper-parameter optimization (HPO) tasks.

The paper in its current form seems incomplete. Ideally, the paper should add a few meaningful real-world domains (as HPO tasks are not convincing to make a strong case) in their empirical evaluation and see what recommendations they can draw. Additionally, another test for the usefulness of the identified recommendations is to show how to apply them in those real-world domains where some of the problem characteristics are not known.

Therefore, I recommend rejecting this paper and strongly encourage the authors' to improve it for a future re-submission.

**Reviewer Concerns:**

Two outstanding concerns:
1. Lack of experiments on real-world domains
2. Applicability of the recommendations when some of the problem characteristics are not known

**Reviewer Scores:**

The first three reviewers' would have likely kept their score same.

Reviewer rckW could have increased the score from 2 to 3 given that statistical tests concern is addressed.

---

### Decision · Program_Chairs · 2026-01-26

Reject